# The nuclear envelope protein Net39 is essential for muscle nuclear integrity and chromatin organization

Andres Ramirez-Martinez [1,2,3,8], Yichi Zhang [1,2,3,8], Kenian Chen [4], Jiwoong Kim [4], Bercin K. Cenik [1,2,3,7], John R. McAnally[1,2,3], Chunyu Cai [5], John M. Shelton [6], Jian Huang[6], Ana Brennan[1,2,3], Bret M. Evers [5], Pradeep P. A. Mammen [2,3,6], Lin Xu [4], Rhonda Bassel-Duby [1,2,3], Ning Liu [1,2,3,9✉] & Eric N. Olson [1,2,3,9✉]

Lamins and transmembrane proteins within the nuclear envelope regulate nuclear structure and chromatin organization. Nuclear envelope transmembrane protein 39 (Net39) is a muscle nuclear envelope protein whose functions in vivo have not been explored. We show that mice lacking Net39 succumb to severe myopathy and juvenile lethality, with concomitant disruption in nuclear integrity, chromatin accessibility, gene expression, and metabolism. These abnormalities resemble those of Emery–Dreifuss muscular dystrophy (EDMD), caused by mutations in A-type lamins (LMNA) and other genes, like Emerin (EMD). We observe that Net39 is downregulated in EDMD patients, implicating Net39 in the pathogenesis of this disorder. Our findings highlight the role of Net39 at the nuclear envelope in maintaining muscle chromatin organization, gene expression and function, and its potential contribution to the molecular etiology of EDMD.

[1] Department of Molecular Biology, University of Texas Southwestern Medical Center, Dallas, TX, USA. [2] Hamon Center for Regenerative Science and Medicine, University of Texas Southwestern Medical Center, Dallas, TX, USA. [3] Senator Paul D. Wellstone Muscular Dystrophy Cooperative Research Center, University of Texas Southwestern Medical Center, Dallas, TX, USA. [4] Department of Population and Data Sciences, Quantitative Biomedical Research Center University of Texas Southwestern Medical Center, Dallas, TX, USA. [5] Department of Pathology, University of Texas Southwestern Medical Center, Dallas, TX, USA. [6] Department of Internal Medicine, University of Texas Southwestern Medical Center, Dallas, TX, USA. [7] Present address: Simpson Querrey Center for Epigenetics, Department of Biochemistry and Molecular Genetics, Northwestern University Feinberg School of Medicine, Chicago, IL, USA. [8] These authors contributed equally: Andres Ramirez-Martinez, Yichi Zhang. [9] These authors jointly supervised this work: Ning Liu, Eric N. Olson. ✉email: Ning.Liu@utsouthwestern.edu; Eric.Olson@utsouthwestern.edu

The nuclear envelope is a double lipid bilayer that separates the cytosol from the nucleoplasm. Beneath the inner nuclear membrane is the nuclear lamina, which includes the intermediate filaments formed by Lamins A, B1, B2, and C, providing support to the nucleus. Lamins and transmembrane proteins within the nuclear envelope are involved in maintaining nuclear envelope structure, nuclear positioning, and chromatin organization[1–5]. The transport of molecules between the nucleus and cytosol is mediated by the nuclear pore complex[6], whereas mechanical communication between both compartments occurs through the linker of nucleoskeleton and the cytoskeleton (LINC) complexes[4,5], comprised of Sad1/UNC84 (Sun) 1 and 2 proteins at the inner nuclear membrane and the nuclear envelope spectrin-repeat proteins (Syne/Nesprins) at the outer nuclear membrane. The LINC complexes act as nuclear bridges to connect the cytoskeleton with the lamin nucleoskeleton and allow mechanical crosstalk between both compartments[4,7,8]. Lamins and nuclear envelope proteins have also been shown to play important roles in genome organization and regulation of gene expression. In eukaryotes, transcriptionally active euchromatin is typically found in the nuclear interior, whereas transcriptionally silent heterochromatin adjoins the nuclear envelope[9–11] and associates with nuclear lamins. Lamin-associated regions of DNA, termed lamin-associated domains (LADs), are dynamic and can redistribute upon gene activation[12,13].

Mutations in nuclear envelope proteins and lamins cause numerous human diseases termed envelopathies, such as Emery–Dreifuss muscular dystrophy (EDMD). EDMD is characterized by skeletal muscle weakness, early contractures, and cardiomyopathy[7,14]. The two most frequent genetic causes of EDMD are X-linked recessive loss of Emerin (encoded by *EMD*) and autosomal-dominant mutations in A-type lamins (encoded by *LMNA*)[15,16]. Additional genes involved in the pathogenesis of EDMD[7,17], including *FHL1*, *TMEM43*, Nesprins, and *SUN2*[18–21], have also been identified.

Proteomic analysis of isolated nuclear envelopes[22] has identified nuclear envelope transmembrane proteins (NETs) with potential links to human disease[23]. Among them, transmembrane protein 39 (Net39), also referred to as inactive phospholipid phosphatase 7 (Plpp7 or Ppapdc3), has been studied in vitro with conflicting results[1,24,25]. Net39 was initially reported to inhibit mammalian target of rapamycin (mTOR) activity and insulin-like growth factor 2 signaling, and knockdown of Net39 in C2C12 cells was shown to strongly promote myoblast differentiation[24]. In contrast, later studies[1,25] demonstrated that knockdown of Net39 blocked myogenesis and the main function of Net39 was to reposition specific genes that inhibit myoblast differentiation to the nuclear periphery, thus repressing their expression.

Here we explored the role of Net39 as a muscle-specific regulator of nuclear envelope structure in vivo by studying a Net39 knockout (KO) mouse model. We show that deletion of Net39 in mice causes profound changes in nuclear envelope integrity, chromatin organization, gene expression, and metabolism, culminating in severe defects in muscle growth and function that lead to juvenile lethality. Furthermore, Net39 is downregulated in EDMD patient biopsies, highlighting the importance of Net39 for proper muscle function and its potential contribution to human disease.

## Results

### Net39 is muscle enriched and induced during myogenesis. The muscle nuclear envelope protein Net39 has been shown to modulate myoblast differentiation in vitro[1,24], but its precise molecular functions within muscle in vivo are unknown. Net39 is upregulated during differentiation of immortalized C2C12 myoblasts, as well as differentiation of sublaminal muscle stem cells marked by the transcription factor Pax7 (also called satellite cells) and interstitial myogenic muscle progenitor cells marked by the transcription factor Twist2 (also called Tw2+ cells)[26] (Supplementary Fig. 1a, b). The expression of Net39 is regulated by the myogenic transcription factors MyoD and Myogenin (Supplementary Fig. 1a, c). In mice, Net39 transcript is enriched in skeletal muscle compared to other tissues. Within the skeletal muscle, Net39 expression is higher in fast-twitch muscles (such as tibialis anterior) than in slow-twitch (soleus) muscles (Supplementary Fig. 1d). Net39 is also expressed in sites of embryonic myogenesis (Supplementary Fig. 1e) and is upregulated during postnatal muscle growth (Supplementary Fig. 1f).

### Loss of Net39 in mice causes juvenile lethality and muscle abnormalities. To explore the functions of Net39 in vivo, we generated Net39 KO mice using CRISPR/Cas9-mediated genome editing. The murine *Net39* gene spans two exons. We deleted the first exon of the gene using two single-guide RNAs (sgRNAs; Supplementary Fig. 2a, b), eliminating most of the open reading frame (ORF). Loss of Net39 mRNA and protein in KO mice was confirmed by RNA sequencing and western blot analysis, respectively, in quadriceps at 17 days postpartum (P17) (Supplementary Fig. 2c, d). Net39 heterozygous mice showed no discernible abnormalities. Net39 KO mice were born at Mendelian ratios from heterozygous intercrosses and were indistinguishable from wild-type (WT) littermates at birth. However, KO mice failed to thrive and were readily identifiable by P7 by their runted appearance, which became increasingly apparent with age (Fig. 1a, b). Net39 KO mice showed progressive lethality starting from P13, with none surviving past 23 days of age (Fig. 1c). KO mice displayed a waddling gait at P17, which was associated with an increasing number of falls, indicative of muscle dysfunction. Histological analysis and wheat germ agglutinin (WGA) staining of skeletal muscles revealed a reduction in myofiber cross-sectional area in KO compared to WT mice at P17 (Fig. 1d, e). We did not observe a decrease in the number of nuclei per fiber or the percentage Pax7+ satellite cells relative to the total number of myonuclei (Supplementary Fig. 3a, b). Isolated extensor digitorum longus (EDL) and soleus muscles from KO mice displayed lower tetanic force (Fig. 1f). Innervation of quadriceps muscle was also assessed by immunofluorescence of neuronal axons (neurofilament, SV2) and acetylcholine receptor (bungarotoxin) in quadriceps to visualize the neuromuscular junction, and no differences were observed between WT and KO muscle (Supplementary Fig. 3c), indicating that the decreases in muscle size and strength in Net39 KO mice are not caused by impaired innervation. Electron microscopy showed sarcomeric disarray in KO diaphragm (Fig. 1g). Aside from defects in the skeletal muscle, we observed that cardiomyocytes in Net39 KO hearts were smaller than WT cardiomyocytes, but this did not lead to a reduction in heart function as measured by echocardiography (ECHO; Supplementary Fig. 4).

### Net39 maintains nuclear integrity and interacts with components of the nuclear envelope. Net39 KO mice showed deformations of nuclear envelope architecture in longitudinal sections of different muscle types and their frequency increased with age (Fig. 2a). Immunohistochemistry for the nuclear envelope protein SUN2 showed that nuclear deformations were characterized by pronounced invaginations and projections that protruded into the sarcomeres (Fig. 2b). Electron microscopy revealed a jagged appearance of KO nuclear envelopes (Fig. 2c). Loss of nuclear envelope proteins like Lamin A/C can also cause nuclear deformations and susceptibility to mechanical stress[27]. Indeed, we

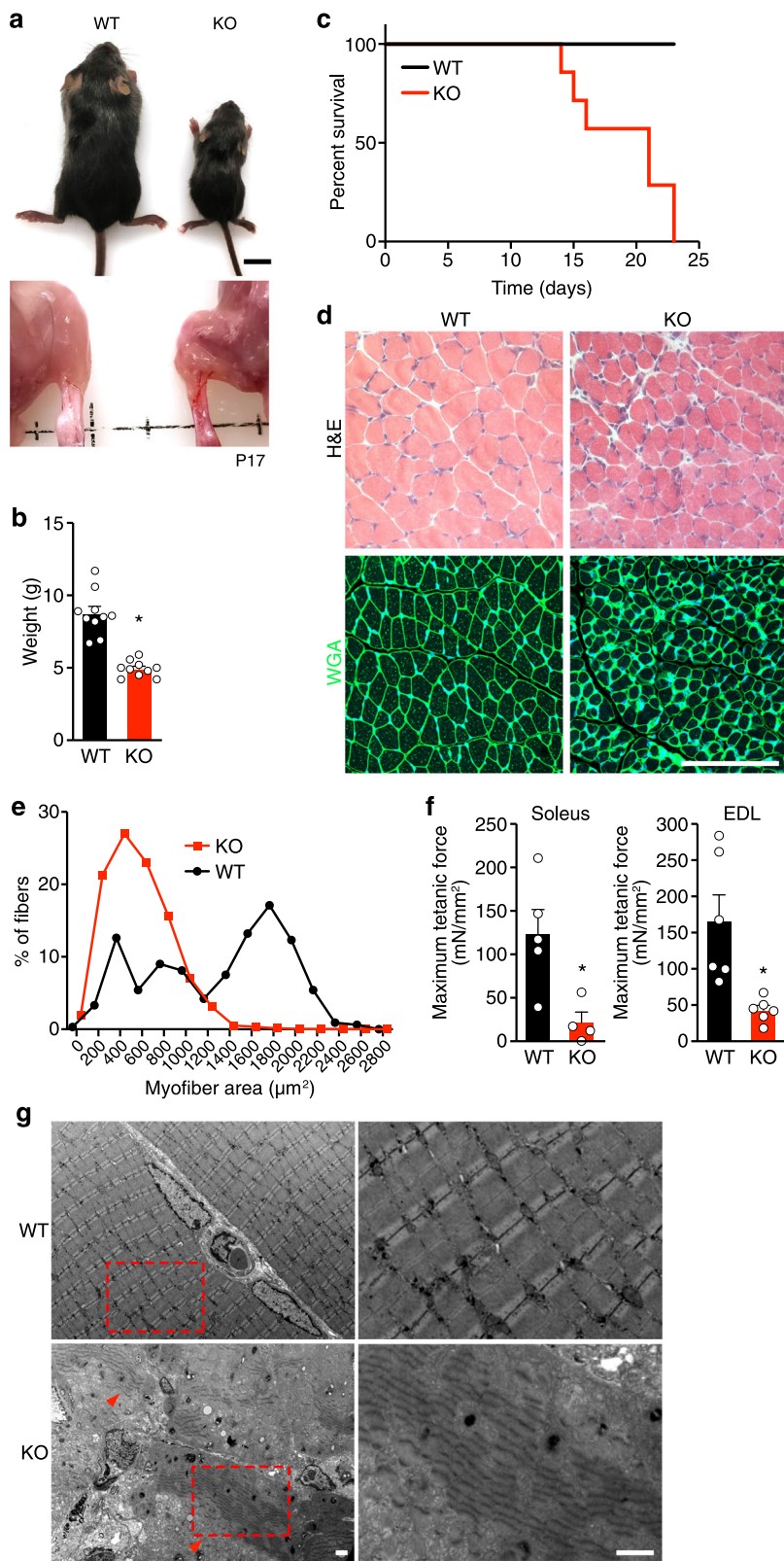

observed that nuclear envelope deformations in KO EDL muscles increased mildly just after a short ex vivo mechanical stretch, whereas WT muscles were unaffected by stretch (Fig. 2d).

To further characterize the nuclear envelope deformations observed in Net39 KO muscles, we probed for changes in nuclear envelope proteins by western blot analysis in Net39 KO muscles and found that LMNB1 and LEMD2 proteins were increased

relative to WT muscle, whereas EMD levels were decreased. These results indicate changes in protein levels or posttranslational changes in the regions recognized by the antibodies used (Fig. 2e, f). Next, we used proximity biotinylation (BioID) to assess whether Net39 is associated with any of these proteins in C2C12 myotubes (Supplementary Fig. 5 and Supplementary Data 1). Indeed, we found that Net39 is associated with multiple

**Fig. 1 Net39 is required for normal muscle structure and function. a** At P17, Net39 KO mice (upper panel) and their hindlimb muscles (lower panel) are abnormally small compared to WT mice. Scale bar: 1 cm. **b** Body weight of WT and Net39 KO mice at P17. $n = 10$ mice per group. * indicates $p = 0.0001$. Data are presented as mean ± SEM values. **c** Survival curve of Net39 KO mice. $n = 8$ mice for WT and 7 mice for KO. **d** H&E (upper panels) and wheat germ agglutinin (WGA) staining (lower panels) of WT and Net39 KO quadriceps muscle at P17. Scale bar: 100 μm. **e** Myofiber area distribution quantified using CellProfiler from WGA-stained quadriceps sections at P17. $p < 0.01$. $n = 3$ mice per group. **f** Ex vivo contraction assay to measure maximum tetanic force of the indicated muscles at P17. EDL extensor digitorum longus. $n = 6$ mice per group for EDL and $n = 4$ for soleus. * indicates $p = 0.0186$ for EDL and $p = 0.0069$ for soleus. Data are presented as mean ± SEM values. **g** Electron micrographs show disorganized sarcomeres in Net39 KO diaphragms at P17. Red arrowheads show sarcomere disarray (left). Higher magnification of the indicated areas (red box) is shown (right). Scale bars: 1 μm. Experiment was performed with three animals per genotype. All statistical comparisons between groups were evaluated by unpaired and two-sided Student's $t$ test. Source data are provided as a Source data file.

components of the nuclear envelope such as LEMD2, SUN2, and EMD (Fig. 2g). In contrast to prior studies[1], we did not observe interaction between Net39 and Lamin A. We further validated the interaction between NET39 and endogenous LEMD2 by co-immunoprecipitation (co-IP; Fig. 2h). Interestingly, a homozygous missense mutation in *LEMD2* that causes arrhythmic cardiomyopathy in humans also leads to nuclear envelope deformations in cardiomyocytes resembling those seen in Net39 KO myonuclei[28], implying a common function. These results suggest that Net39 associates with multiple components of the nuclear envelope and loss of Net39 compromises nuclear envelope integrity.

**Net39 modulates genome organization and gene expression.** Nuclear envelope proteins regulate the formation of genomic regions associated with lamins called LADs[12,13]. We sought to profile global changes in LADs by performing Lamin A/C chromatin immunoprecipitation–sequencing (ChIP-seq) in WT and Net39 KO muscles. We found that there was a significant loss of LADs and transcriptional start sites within the LADs in Net39 KO quadriceps muscles (Supplementary Fig. 6a, b). We also assessed chromatin accessibility in WT and Net39 KO quadriceps using assay for transposase-accessible chromatin sequencing (ATAC-seq; Fig. 3a). Following Net39 deletion, we observed differential changes in chromatin accessibility in genes related to distinct pathways (Fig. 3b). The promoters of genes involved in lipid metabolism such as Acyl-CoA transferase 1 (*Acot1*) became more accessible (Supplementary Fig. 6c), whereas genes related to carbohydrate metabolism and muscle contraction such as fast fiber-type myosin-binding protein C (*Mybpc2*) became less accessible (Supplementary Fig. 6d).

Changes in gene expression were profiled by RNA sequencing at early (P9) and late (P17) time points. Differences between KO and WT quadriceps were more pronounced over time (Fig. 3c). Differentially regulated genes correlated with the changes in their accessibility (Fig. 3d). An example of this correlation is Cysteine and glycine-rich protein 3 (*Csrp3*), also known as muscle LIM protein, an inhibitor of myoblast differentiation[29]. *Csrp3* expression is affected by the loss of a LAD, which may facilitate chromatin opening and increased accessibility of its gene body, resulting in a concomitant increase in its mRNA expression (Supplementary Fig. 6e). Pathway analysis showed that the upregulated genes in KO muscle, such as *Acot1* and *Csrp3*, were mainly involved in lipid and muscle processes (Fig. 3e and Supplementary Fig. 7a, b). In contrast, the downregulated genes were related to carbohydrate metabolism, cell cycle, and mitotic division (Fig. 3e and Supplementary Fig. 7c, d). Downregulated cell cycle genes such as *Cdk1* and *Aurkb* control nuclear envelope assembly and disassembly by regulating chromosomal condensation[30], whereas kinesins (*Kif4*, *Kif22*) have been shown to be required in the muscle for nuclear positioning[31]. We observed alterations in the expression of myosin heavy chain genes and key sarcomeric components, including myozenin-1, myosin-binding

protein C, myosin light chain 1, and tropomyosin 1 (Supplementary Fig. 7e, f), which likely contributes to sarcomere disarray observed in the diaphragm (Fig. 1g).

**Transcriptional changes in Net39 KO mice affect muscle fiber type and metabolism.** To further explore the functional consequences of transcriptional alterations, we analyzed the fiber-type composition of KO muscle. Immunohistochemical staining showed that Net39 KO quadriceps exhibited a shift toward an oxidative fiber composition (type I and type IIa; Supplementary Fig. 8a, b). An increase in oxidative metabolism was also evidenced by NADH, succinic dehydrogenase (SDH), and cytochrome oxidase (COX) staining on quadriceps muscle (Supplementary Fig. 8c). Consistent with these results, functional analysis of EDL muscle revealed an increase in fatigue resistance (Supplementary Fig. 8d). To determine whether the increase in oxidative metabolism is a result of increased mitochondria biogenesis, we examined the mitochondrial DNA content by quantitative polymerase chain reaction (qPCR) in quadriceps muscle samples and observed no differences between WT and Net39 KO mice (Supplementary Fig. 8e). Consistently, the transcript levels of the regulator of mitochondrial biogenesis PGC1α did not change (Supplementary Fig. 8f). Therefore, we conclude that the increased oxidative metabolism in Net39 KO muscle is not due to an increased number of mitochondria. While mTOR has been shown to be regulated by Net39 in transfected HeLa cells[24], no changes in mTOR signaling were observed in Net39 KO muscles (Supplementary Fig. 8g) and Net39 overexpression in C2C12 cells did not change mTOR localization (Supplementary Fig. 8h).

To better understand the changes in metabolism, we performed targeted metabolomics on WT and Net39 KO hindlimb muscle (Supplementary Fig. 9a, b and Supplementary Data 2) and observed increases in fatty acid species and decreases in glycolysis intermediates (Supplementary Fig. 9c, d). Overall, Net39 deletion caused a shift from carbohydrate to lipid metabolism in the skeletal muscle, which may contribute to increased oxidative activity[32]. Serum glucose levels were significantly decreased in Net39 KO mice compared to WT, but no changes were observed in other serum metabolites or circulating insulin (Supplementary Fig. 9e). Hypoglycemia has also been observed in mouse models of EDMD, in which *Lmna* is either deleted or mutated[33,34].

**Net39 is downregulated in EDMD.** Several human mutations in *NET39* have been identified as novel alleles associated with EDMD[23]. EDMD is a complex disease and there are multiple mouse models that reflect its genetic diversity and broad range of manifestations[7]. The early lethality and nuclear envelope dysregulation observed in Net39 KO mice prompted us to compare the Net39 KO phenotype with other mouse models of severe EDMD. One such model carries a Lmna ΔK32 mutation, a single amino acid deletion that impairs the lateral assembly of Lamin A/C[35]. In humans, the Lmna ΔK32 mutation causes severe congenital

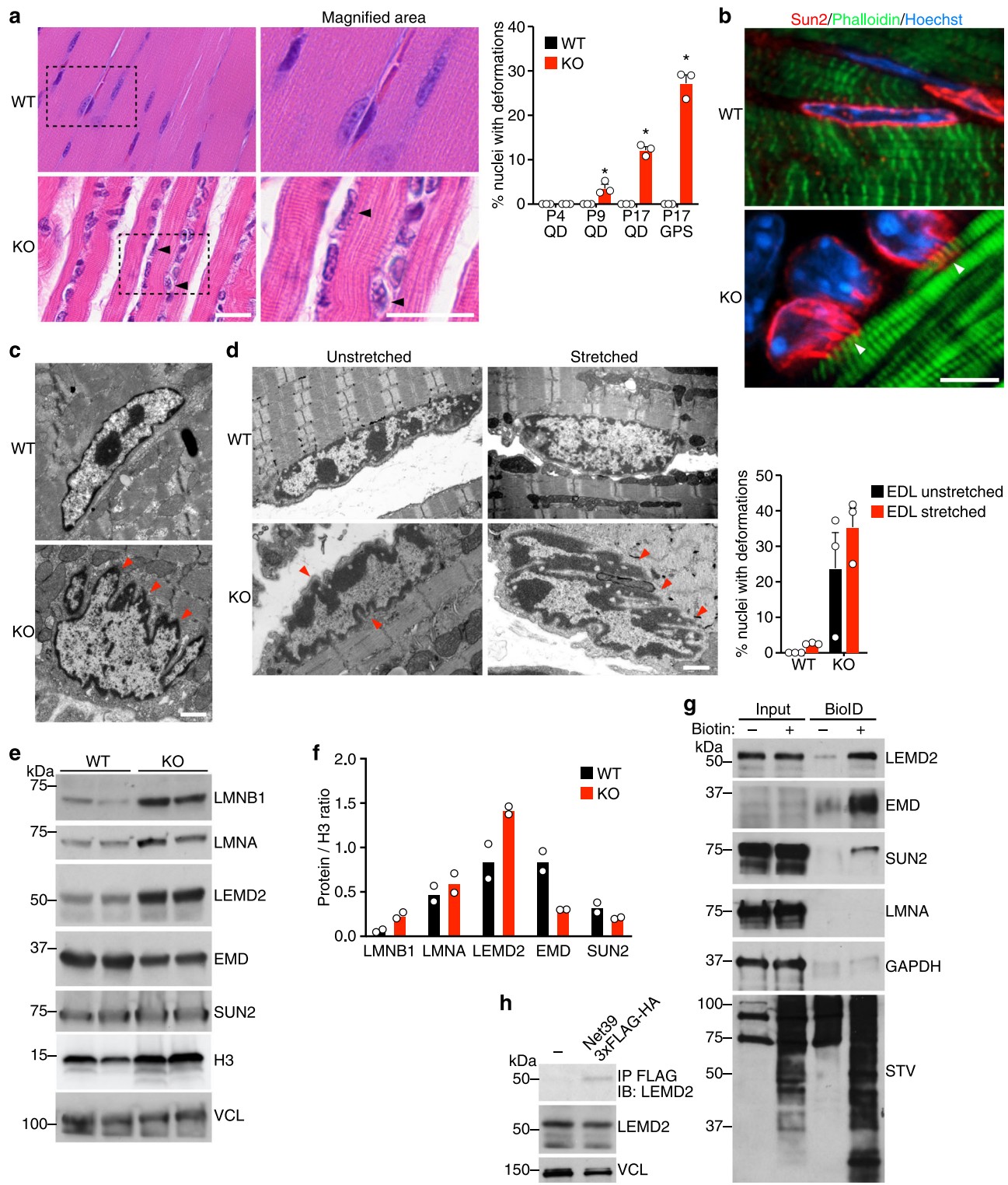

muscular dystrophy[36]. We observed overlapping phenotypes between Lmna ΔK32 mice[33] and our Net39 KO mice. Both mouse models manifest early lethality, nuclear abnormalities, failure to grow, and metabolic alterations. Furthermore, it was recently reported that Lmna ΔK32 myotubes show down-regulation of Net39 transcript and protein levels[37]. These findings raised the possibility that Net39 expression may be affected in EDMD patients and may contribute to the pathogenesis of the disease.

To determine whether Net39 is downregulated in EDMD, we examined Net39 expression in muscle biopsies from patients with EDMD caused by different missense mutations in the *LMNA* gene (Fig. 4a and Table 1). We found that NET39 protein levels were decreased by >80% in these muscles (Fig. 4b, c), with a concomitant decrease in NET39 transcript (Fig. 4d). In contrast, Net39 expression was unaffected in a mouse model of Duchenne muscular dystrophy (DMD), caused by deletion of exon 44 (ΔEx44) of the *Dmd* gene[38] (Supplementary Fig. 10). Protein levels of another

**Fig. 2 Net39 maintains integrity of the nuclear envelope. a** H&E staining of longitudinal quadriceps sections at P17 (left). Magnified images display the jagged outlines of the indicated nuclei. Arrowheads indicate nuclear envelope protrusions. Scale bars: 20 μm. Quantification of the percentage of nuclei with nuclear envelope deformations in the indicated muscles and time points (right). $n = 3$ WT and KO mice. For each analysis, P9 QD * indicates $p = 0.0449$, P17 QD $p = 0.0037$ P17 GPS $p = 0.0043$. QD quadriceps, GPS gastrocnemius plantaris soleus. **b** Immunofluorescence of the nuclear envelope in longitudinal quadriceps sections at P17 reveals nuclear envelope deformations. Sections were stained for the inner nuclear membrane protein Sun2 (red), phalloidin for F-actin (green), and Hoechst for DNA (blue). Arrowheads indicate nuclear envelope protrusions. Scale bar: 5 μm. Experiment was performed with two animals per genotype. **c** Electron micrographs of P17 quadriceps nuclei showing nuclear envelope defects in Net39 KO muscle. Arrowheads indicate nuclear envelope protrusions. Scale bar: 1 μm. Experiment was performed with three animals per genotype. **d** Electron micrographs of P17 extensor digitorum longus (EDL) nuclei before and after ex vivo stretching (left) and quantification of the percentage of nuclei with nuclear envelope deformations (right). Arrowheads indicate nuclear envelope protrusions. Scale bar: 1 μm. $p = 0.3$. $n = 3$ WT and KO mice. Data are presented as mean ± SEM values. **e** Western blot analysis showing protein levels of LMNB1, LMNA, LEMD2, EMD, and SUN2 in P17 quadriceps muscle lysates from WT and Net39 KO mice. Vinculin (VCL) and Histone H3 are loading controls for total protein and nuclear protein, respectively. **f** Densitometry quantification of western blot from Fig. 2e. The intensity of each nuclear envelope protein was normalized to Histone H3 intensity. $n = 2$ biological replicates. **g** Net39 BioID in C2C12 myotubes detects enrichment of biotinylated LEMD2, SUN2, and EMD but not Lamin A in C2C12 myotubes. GAPDH is a loading control. Total biotinylated proteins were detected using streptavidin-HRP (STV). Two independent experiments were performed. **h** Validation of Net39 interaction with Lemd2 by co-immunoprecipitation in C2C12 myotubes. VCL is a loading control. Two independent experiments were performed. All statistical comparisons between groups were evaluated by unpaired and two-sided Student's $t$ test. Source data are provided as a Source data file.

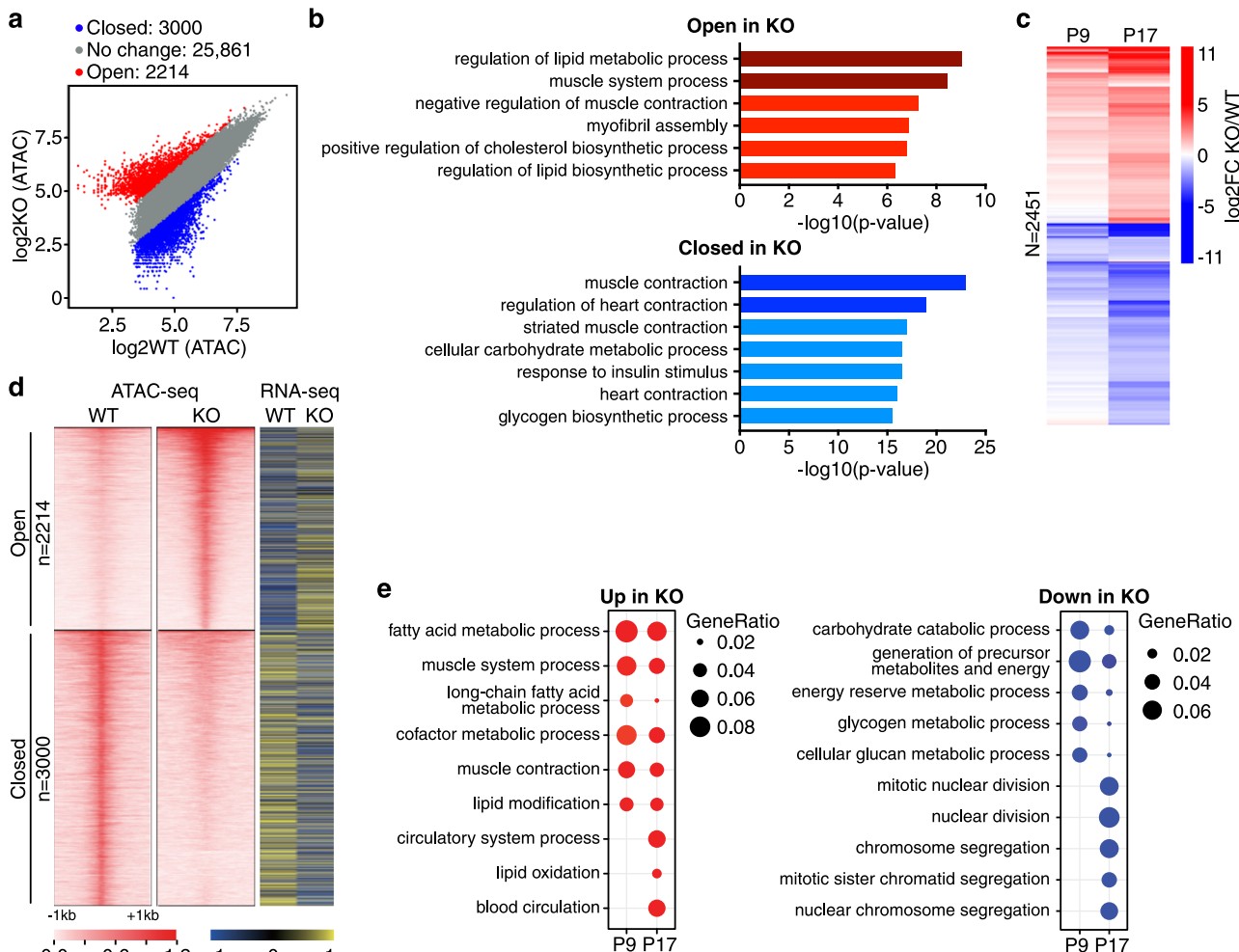

**Fig. 3 Loss of Net39 causes changes in chromatin accessibility and gene expression. a** Scatter plot with all detected peaks in P17 quadriceps ATAC-seq ($n = 3$). A cutoff of fold change >2 and an adjusted $p$ value <0.05 by Benjamini and Hochberg procedure for multiple comparisons was set for the identification of differentially open and closed peaks in Net39 KO samples. **b** Pathways enriched by GREAT analysis of open or closed chromatin peaks in Net39 KO samples compared to WT. **c** Heatmap showing fold change of upregulated and downregulated genes in KO muscle at P9 ($n = 3$) and P17 ($n = 3$) as determined by RNA-seq. A cutoff of fold change >2 and an adjusted $p$ value <0.05 by Benjamini and Hochberg procedure for multiple comparisons was set for the identification of differentially expressed genes. Color scale represents $Z$-score. **d** Heatmap of open- and closed-regulated ATAC-seq peaks (left panels) and the expression level of associated genes as determined by RNA-seq (right panel). Color scale represents $Z$-score. **e** GO terms enriched among upregulated (left) and downregulated (right) genes by RNA-seq. Circle sizes indicate the ratio of genes included in the pathway.

muscle-enriched nuclear envelope protein, LEMD2, were unchanged in EDMD patients (Fig. 4b, c). Taken together, these results highlight the potential contribution of Net39 to EDMD.

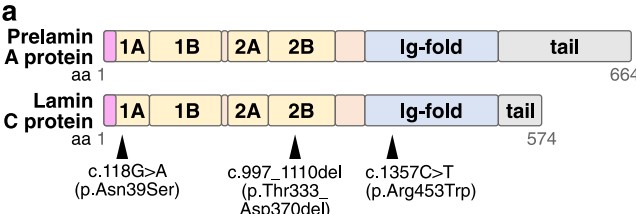

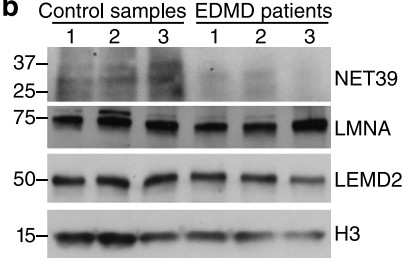

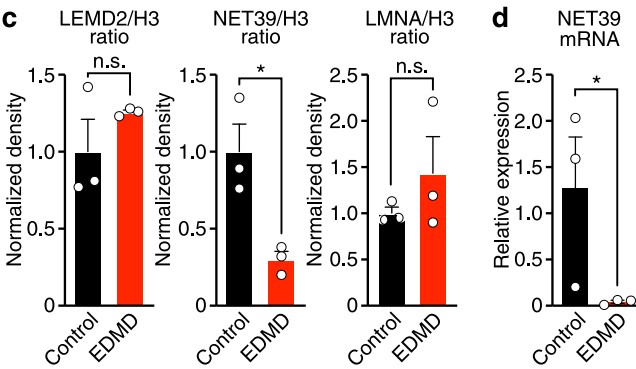

**Fig. 4 Net39 is downregulated in EDMD. a** Illustration of Lamin A/C protein domains highlighting specific mutations in three Emery–Dreifuss muscular dystrophy (EDMD) patients. In the Lamin A/C protein domains, pink indicates head region, yellow denotes rod domains, blue shows Ig fold, and gray indicates the tail regions of Lamin C and pre-Lamin A. **b** Western blot analysis of human muscle biopsies from healthy (control) individuals and EDMD patients showing NET39, LMNA, and LEMD2 protein expression. Histone H3 (H3) was used as a nuclear loading control. **c** Densitometry of western blots shown in **b** was performed and the ratios of LEMD2/H3 ($p = 0.2903$) (left), NET39/H3 (middle) (* indicates $p = 0.0199$), and LMNA/H3 (right) ($p = 0.3454$) were quantified. $n = 3$ biologically independent samples. Data are presented as mean ± SEM values. Statistical comparisons between groups were evaluated by unpaired and two-sided Student's $t$ test. **d** qRT-PCR analysis of Net39 transcript in control and EDMD patient biopsy samples. Net39 mRNA expression in patient samples is normalized to its expression in control samples. * indicates $p = 0.0445$. $n = 3$ biologically independent samples. Data are presented as mean ± SEM values. Statistical comparisons between groups were evaluated by unpaired, one-tailed Student's $t$ test. Source data are provided as a Source data file.

## Discussion

Our findings reveal an essential role for Net39 in muscle growth and function. The loss of Net39 leads to compromised nuclear envelope integrity in addition to changes in chromatin organization, gene expression, and muscle metabolism. Mice lacking Net39 fail to thrive and display reduced muscle growth and function. These characteristics resemble an EDMD-like phenotype in mouse models with Lamin A/C deletion or mutations[33,39,40], and the nuclear envelope of Lamin A/C-deficient fibroblasts displays similar morphological abnormalities in response to stretch[27].

In vitro, Net39 has been described as both a negative and a positive regulator of C2C12 myoblast differentiation by regulating mTOR signaling and gene positioning, respectively[1,24]. However, we did not observe changes in mTOR signaling in Net39 KO mice or in C2C12 cells overexpressing Net39. It is possible that the previously reported functions may be specific to transfected HeLa cells and not observed in muscle cells. In contrast, Net39 KO mice presented extensive changes in genome organization that compromised muscle function. These observations are consistent with the other proposed function of Net39: to regulate myoblast differentiation by sequestering repressors of myogenesis to the nuclear periphery[1].

Unbiased identification of Net39 interactors by proximity labeling (BioID) showed different results from those reported previously. By myc-Net39 pulldown, the nucleoplasmic N-terminus of Net39 was previously shown to interact with mTOR[24], and the N-terminus of Net39 has also been proposed to associate with Lamin A/C for genome tethering[1]. Our proximity labeling data showed association of Net39 with EDMD-related proteins but not with mTOR or any lamins. One explanation for the differences is that Net39 may control genome organization by interacting with lamin-associated proteins rather than lamins themselves. It is also possible that the interaction with lamins may be transient or too weak to be detected by BioID. Alternatively, the N-terminus of Net39 may not be accessible to the biotin ligase fused to the C-terminus of Net39 used in this study. In contrast, we observed by proximity labeling and co-IP that Net39 interacts with Lemd2, and loss of Net39 results in upregulation of Lemd2 transcript and protein levels. Nuclear envelope deformations like those observed in Net39 KO mice have also been reported in cardiomyopathy patients carrying a mutation in *LEMD2*[28]. Lemd2 tethers chromatin and Lamin A/C to the nuclear periphery through its nucleoplasmic LEM domain[41]. We hypothesize that both proteins may potentially act as part of a complex to regulate nuclear organization, transcription, and mechanical properties of the envelope. The upregulation of Lemd2 expression in Net39 KO muscle is likely a compensatory mechanism.

Generating a Net39 KO mouse model allowed us to understand the role of a tissue-specific nuclear envelope protein in vivo. Net39 expression is restricted to skeletal muscle and deletion of Net39 caused more nuclear deformations in the gastrocnemius plantaris soleus muscles than in the quadriceps muscle. Muscle groups can be differentially affected under pathological conditions, such as muscular dystrophy[42]. We propose that heterogeneity in Net39 expression (enriched in fast-twitch muscle), mechanical burden, and fiber-type

**Table 1 Human sample information.**

| Group | Collection site | Age range (mean), years | *LMNA* mutations |
|---|---|---|---|
| Healthy | Quadriceps or deltoid | 3–10 (6.3) | None |
| EDMD | Thigh or quadriceps | 1.5–3 (2.5) | c.1357C>T, p.Arg453Trp, heterozygous |
| | | | c.116G>A, p.Asn39Ser, heterozygous |
| | | | c.997_1110del, p.Thr333_Asp370del, heterozygous |

composition may account for subtle phenotypical differences across muscles. With an in vivo model, we were also able to examine the contribution of Net39 to muscle metabolism. Net39 KO mice presented a fiber-type switch toward oxidative fibers with concomitant changes in contractility and metabolism but no changes in total mitochondrial number. Those metabolic changes are similar to those described for other envelopathies[33,34]. Further mechanistic studies should be performed to understand how Net39 controls multiple processes in muscle, including growth, metabolism, and contractility.

We showed that Net39 expression is diminished in EDMD patients with dominant LMNA missense mutations. Specifically, Net39 protein and transcript levels are decreased in muscle biopsies from EDMD patients relative to healthy controls but not in a mouse model of muscular dystrophy (DMD). Mutations in LMNA have been proposed to impair MyoD activation in the context of EDMD[43], and Net39 is a MyoD target (Supplementary Fig. 1c). Mutations in LMNA can also influence Net39 levels[37]. We surmise that LMNA-dependent MyoD dysregulation could underlie the loss of the muscle-specific nuclear envelope protein Net39 in EDMD. Reduced Net39 levels may contribute to the muscle defects observed in EDMD. It will be of interest to understand the role of Net39 in this disease and the transcriptional similarities and differences with other models for EDMD. Overall, our findings show that loss of the nuclear envelope protein Net39 causes profound defects in mice, and the reduced Net39 levels in EDMD patients potentially contribute to the pathogenesis of this disorder.

## Methods

**Generation of Net39 KO mice.** All animal procedures were approved by the Institutional Animal Care and Use Committee at the University of Texas Southwestern Medical Center.

CRISPR Cas9 guides flanking exon 1 of the Net39 (also referred to as Plpp7) gene were selected from CRISPR 10K Genome Browser Track, cloned into pX458 (Addgene # 251928), transfected into N2a cells, FACS sorted, and cutting efficiency was assessed by T7E1 assay as per the provider's instructions (New England BioLabs #E3321).

#1Net39-sgRNA-5'5'-TCCCTGAACCAGCCCCCCAA-3'
#2Net39-sgRNA-3'5'-GGGGTTGGGGCCGGCTCCCAGA-3'

Cas9 mRNA and Net39 sgRNAs (#1 and #2) were injected into the pronucleus and cytoplasm of zygotes. For zygote production, B6C3F1 female mice were treated for superovulation and mated to B6C3F1 stud males. Zygotes were isolated, transferred to M16 and M2 medium, injected with Cas9 mRNA and sgRNA, and cultured in M16 medium for 1 h at 37 °C. Injected zygotes were transferred into the oviducts of pseudo-pregnant ICR female mice.

Tail genomic DNA was extracted from F0 mice and used for genomic analysis with PCR primers that amplify the targeted region. Primers 1 and 2 amplify fragments of different size in WT and KO mice. Primers 2 and 3 only amplify the WT allele.

#1Net39-WT/KO-F5'-GCAGCTGGAGGTAAATAGCC-3'
#2Net39-WT/KO-R5'-CTCCCCACACTAGAGGCTTG-3'
#3Net39-WT.only-F5'-GCAGATGTCAATAGCCAGCA-3'

Mosaic mice were mated to C57BL6N mice and a mouse line with a 559-bp deletion was selected for further characterization. Experiments requiring mice were sex balanced (2 males, 1 female per genotype).

**Radioisotopic in situ hybridization (ISH).** Radioisotopic ISH was performed on E12.5 embryo sections and modified from prior protocols[44]. For pre-hybridization, embryo section slides were heated to 58 °C for 30 min, deparaffinized in xylene, and hydrated by sequential ethanol/diethyl pyrocarbonate (DEPC)-saline washes (95, 85, 60, 30%) to DEPC-saline. Microwave RNA retrieval was performed in plastic containers (Miles Tissue-Tek, Elkhart, IN) filled with DEPC-1× Antigen Retrieval Citra pH 6.0 (Biogenex, San Ramon, CA), and samples were heated in a 750-watt microwave at 90% power for 5 min. Evaporated solution was replaced with DEPC-H$_2$O, and an additional heating was performed at 60% power for 5 min. Samples were cooled down for 20 min and washed twice in DEPC-phosphate-buffered saline (PBS) for 5 min. Samples were then permeabilized for 7.5 min with 20 μg/ml pronase-E in 50 mM Tris-HCl, pH 8.0, 5 mM EDTA, pH 8.0 in DEPC-H$_2$O. Samples were then washed in DEPC-PBS twice and re-fixed in 4% paraformaldehyde/DEPC-PBS, pH 7.4, for 5 min, washed in DEPC-PBS, and acetylated in 0.25% acetic anhydride/0.1 M triethanolamine–HCl, pH 7.5, twice for 5 min. Slides were then equilibrated in 1× SSC, pH 7.0, for 5 min, incubated in 50 mM n-

ethylmaleimide/1× SSC, pH 7.0, for 20 min, and washed in DEPC-PBS, pH 7.4, and DEPC-saline. Finally, slides were dehydrated through graded ethanol/DEPC-saline rinses (30, 60, 85, 95%) to absolute ethanol and dried under vacuum for 2 h.

Net39 mRNA probe sequence was synthesized by Integrated DNA Technologies (IDT) and cloned into pCRII-Topo vector (ThermoFisher, K460001) as per the provider's instructions. MAXIscript SP6/T7 (Life Technologies, AM1320) was used for in vitro transcription of the probe. The following sequence was used as probe:

CCTGCTGGCTATTGACATCTGCATGTCCAAGCGACTGGGGGTGTGTG CCGGCCGGGCTGCATCCTGGGCCAGCGCCCGCTCCATGGTCAAGCTCAT TGGCATCACAGGCCACGGCATTCCTTGGATCGGGGGCACCATCCTCTGC CTGGTGAGAAGGCAGCACCCTGGCTGGCCAAGAGGTGCTCATGAACCTGC TGCTAGCCCTGCTCTTGGACATCATGACAGTGGCTGGAGTCCAGAAGCT CATCAAGCGCCGCGGGCCCATATGAGACCAGCCCTGGGCTCCTGGACTAC CTCACCATGGACATCTATGCCTTCCCTGCCGGCCACGCCAGCCGTGCCG CCATGGTGT

The probe was diluted in hybridization mixture (50% formamide, 0.75 M NaCl, 20 mM Tris-HCl, pH 8.0, 5 mM EDTA, pH 8.0, 10 mM NaPO$_4$, pH 8.0, 10% dextran sulfate, 1× Denhardt's, and 0.5 mg/ml tRNA) to achieve $7.5 \times 10^3$ cpm/μl and heated to 95 °C for 5 min. Diluted probe was cooled to 37 °C and dithiothreitol (DTT) was added to a final concentration of 10 mM. The probe was applied over the sections in a Nalgene Nalgene utility box lined with 5× SSC/50% formamide-saturated gel blot paper. Hybridization was performed for 14 h at 70 °C.

After hybridization, slides were washed as follows: 5× SSC/10 mM DTT at 55 °C for 40 min, HS (2× SSC/50% formamide/100 mM DTT) at 65 °C for 40 min, three 10-min washes in NTE (0.5 M NaCl/10 mM Tris-HCl, pH 8.0/5 mM EDTA, pH 8.0) at 37 °C, NTE with RNase-A (2 μg/ml) at 37 °C for 30 min, NTE at 37 °C for 15 min, HS at 65° for 30 min, 2 washes in 2× SSC and 0.1× SSC each at 37 °C for 15 min. Slides were then dehydrated in graded ethanol rinses (30, 60, 85, 95%) to absolute ethanol and dried under vacuum.

For autoradiographic exposure, dried slides were added pre-warmed (42 °C) diluted Ilford K.5 nuclear emulsion (Polysciences, Warrington, PA) and dried at room temperature at 75% humidity for 3 h. Slides were then sealed with desiccant and stored at 4 °C for 28 days. After that, samples were developed in D19 (Eastman Kodak, Rochester, NY) at 14 °C, and the latent image was fixed with Kodak Fixer. After rinsing, hematoxylin counter-staining was performed (Richard-Allen, Kalamazoo, MI), and samples were dehydrated and mounted. Net39 expression was observed with a Leitz Laborlux-S microscope stand equipped with Plan-EF optics, a standard bright-field condenser, and a Mears low-magnification dark-field condenser.

**Ex vivo electrophysiology.** EDL and soleus muscles were isolated from postnatal day 17 (P17) mice and mounted on Grass FT03.C force transducers connected to a Powerlab 8/SP data acquisition unit (AD Instruments), under physiological salt solution at 37 °C, and continuous flux of 95% O$_2$–5% CO$_2$. Muscles were adjusted to initial length at which the passive force was 0.5 g and then stimulated with two platinum wire electrodes to establish optimal length for obtaining maximal isometric tetanic tension. Measurements were normalized to specific force (mN/mm$^2$) to account for tissue cross-sectional area. For fatigue assays, after reaching optimal muscle length, muscles were stimulated for 20 s (fatigability measurements) or 10 min (for electron microscopy) at 0.8 Hz, 50 ms. Fatigue curves were calculated by comparing the relative change in force to the initial peak (considered 100%).

**Histology, immunochemistry, and electron microscopy.** Skeletal muscle tissues were flash-frozen in a cryoprotective 3:1 mixture of tissue-freezing medium (Triangle BioSciences International) and gum tragacanth (Sigma, G1128) and sectioned on a cryostat and routine hematoxylin and eosin staining was performed. For SDH staining, unfixed frozen sections were incubated in 0.2 M phosphate buffer (pH 7.6) containing sodium succinate and nitroblue tetrazolium chloride (NBT) for 60 min at 37 °C (21). For NADH staining, unfixed frozen sections were incubated in 0.05 M Tris buffer (pH 7.6) containing NADH and NBT for 30 min at 37 °C[45]. Sections were cleared with acetone and mounted in aqueous medium. COX activity was detected in unfixed frozen sections by incubation for 1 h at room temperature in 1 mg/ml cytochrome C (Sigma, C2506)/6 mg/ml catalase (Sigma, C40)/0.5 mg/ml 3,3-diamonobenzidine tetrachloride (Sigma, D5637) in PBS, pH 7.4. Upon conclusion of incubation, sections were washed in distilled water, dehydrated, cleared, and coverslips mounted with synthetic mounting medium (ThermoFisher, SP15-100).

For immunofluorescence, cryosections were fixed in 4% paraformaldehyde for 15 min and permeabilized in 0.3% Triton X-100 for 15 min. Sections were blocked with mouse on mouse blocking solution (Vector Labs, BMK-2202) and 5% goat serum. Primary and conjugated Alexa Fluor secondary antibodies (ThermoFisher) were used at 1:200 dilution. The following antibodies and conjugated fluorophores were used: SUN2 (Sigma, MABT880), Lamin A (Abcam, ab26300), LEMD2 (Sigma, HPA017340), MYH7 (Santa Cruz, sc-53089), MYH2 (Santa Cruz, sc-53096), MYH4 (Proteintech, 20140-1-AP), PAX7 (DSHB, PAX7-c), SV2 (DSHB, SV2-c), Neurofilament (DSHB, 2H3-c), α-Bungarotoxin-555 (ThermoFisher, B35451), Phalloidin-488 (ThermoFisher, A12379), WGA-488 (ThermoFisher, W11261). Confocal images were obtained in Zeiss LSM 800. Myofiber diameter was measured with CellProfiler.

For electron microscopy of the whole muscle, mice were perfused with 4% paraformaldehyde and 1% glutaraldehyde in 0.1 M sodium cacodylate buffer (pH 7.4) and stained with 1% osmium tetroxide. For electron microscopy after ex vivo stretching, muscles were immediately fixed after 10 min of stimulation and contraction. Samples were processed by the University of Texas Southwestern Medical Center Electron Microscopy Core facility. Images were acquired using a FEI Tecnai G2 Spirit transmission electron microscope.

**Plasmids and cloning**. The ORF for Net39 was obtained in pCMV6-Entry backbone (Origene, MR203615). Net39 ORF was subcloned into the following custom-made pMXs-puro (Cell Biolabs, RTV-012) backbones: pMXs-puro-3xFLAG-HA (C-terminal) and pMXs-puro-miniTurbo (C-terminal). pMXs-puro-miniTurbo was generated by conventional cloning using pcDNA-V5-miniTurbo-NES (Addgene, #107170) as the PCR template.

**Cell culture, overexpression, and immunofluorescence**. C2C12 mouse myoblasts, N2a neuroblastoma cells, HEK 293T cells, and Platinum E cells (Cell Biolabs, NC0066908) were cultured in 10% fetal bovine serum with 1% penicillin/streptomycin in Dulbecco's Modified Eagle Medium (DMEM). Cells were transfected with FuGENE6 (Promega, #E2692) as per the provider's instructions. Ten micrograms of plasmid was used for 10-cm plate transfection and 20 μg of DNA was used for 15-cm plate transfection. Platinum E cells were used for retroviral virus production. Forty-eight hours after transfection, supernatants were collected and filtered through a 0.45-μm syringe filter. Virus was concentrated with Retro-X concentrator (Takara, 631456). After 16 h of viral concentration, viral soup was centrifuged at $1500 \times g$ for 45 min, and the pellet was resuspended in fresh growth media supplemented with polybrene (Sigma, H9268) at a final concentration of 8 μg/ml. Twenty-four hours after infection, cells had their media replaced with fresh growth media.

For immunofluorescence, C2C12 cells overexpressing pMXs-puro-Net39-3xFLAG-HA or empty pMXs-puro-3XFLAG-HA were differentiated into myotubes for 5 days, fixed in 4% paraformaldehyde for 15 min, and permeabilized in 0.3% Triton X-100 for 15 min. Cells were blocked with 5% bovine serum albumin (BSA) in PBS for 1 h and incubated in primary and secondary antibodies in blocking solution for 1 h. The following antibodies were used at 1:200 dilution: mTOR (Cell Signaling, 2983) and My32 (Sigma, M4276).

**Western blot analysis and co-IP**. Protein was isolated from flash-frozen muscle samples by addition of RIPA buffer (Sigma, R0278) and mechanical homogenization in Precellys Evolution (3× 20 s at 6800 rpm). Protein concentration was determined by BCA assay (ThermoFisher, 23225), and equal amounts of protein among samples were used for regular western blot and transfer in polyvinylidene fluoride membrane (Millipore, IPVH00010).

For co-IP, stable cell lines of C2C12 myoblasts were generated by infection with retroviral pMXs-puro-Net39-3xFLAG-HA or empty pMXs-puro-3XFLAG-HA. For each condition five 15 cm plates were differentiated into myotubes for 5 days and processed for co-IP. Briefly, cells were washed with PBS and scraped on PBS. Cells were centrifuged at $500 \times g$ for 5 min, and pellets were resuspended in 1.5 ml of 50 mM Tris, 150 mM NaCl, and 0.2% Triton, supplemented with protease inhibitor cocktail (Sigma, 11697498001) and PhosSTOP phosphatase inhibitor cocktail (Sigma, 4906845001). FLAG pulldown and elution were performed using Anti-FLAG M2 Magnetic Beads (Sigma, M8823) and 3xFLAG peptide (Sigma, F4799) as per the provider's instructions. Beads were washed with 50 mM Tris, 350 mM NaCl, and 0.2% Triton before elution. In all, 2% of the lysate was loaded for input and the rest was used for co-IP.

Blocking and antibody incubation were performed in 5% milk in TBS-Tween 0.1%. The following antibodies were used at a 1:200 concentration (primary) or 1:5000 (horseradish peroxidase (HRP) conjugated): NET39 (Sigma, HPA070252), LEMD2 (Sigma, HPA017340), SUN2 (Sigma, MABT880), Lamin A (Abcam, ab26300), LMNB1 (Abcam, ab16048), EMD (Santa Cruz, sc-25284), VCL (Sigma, V9131), GAPDH (Sigma, MAB374), TUBB (Abcam, ab6046), Histone H3 (Cell Signaling, 9715 s), HRP-conjugated streptavidin (Thermofisher, N100), goat anti-Mouse IgG (H + L)-HRP Conjugate (Bio-Rad, 170-6516), and goat anti-Rabbit IgG (H + L)-HRP Conjugate (Bio-Rad, 170-6515). For mTOR signaling, western blot analysis was performed using 5% BSA for blocking and antibody incubation. The following antibodies were used at 1:200 concentration: AKT (Cell Signaling, 9272), RAPTOR (Cell Signaling, 2280), S6K (Cell Signaling, 9202), p-S6K (Thr389) (Cell Signaling, 9234), 4EBP1 (Cell Signaling, 9452), and p-4EBP1 (Ser65) (Cell Signaling, 9451). Immunodetection was performed using Western Blotting Luminol Reagent (Santa Cruz Biotechnology, sc2048).

**Gene expression analysis**. Flash-frozen quadriceps muscle samples from P9 and P17 mice were homogenized in 1 ml of Trizol in Precellys Evolution (3× 20 s at 6800 rpm). RNA was isolated using the RNeasy Micro Kit (Qiagen, 74004) as per the provider's instructions. cDNA was synthesized using iScript Reverse Transcriptase (Bio-Rad).

RNA-seq ($n = 3$ mice per genotype) was performed by the UT Southwestern Genomic and Microarray Core Facility. Single-end raw reads with >30% nucleotide with phred quality scores <20 were filtered from further analysis. Quality-filtered reads were then aligned to the mouse reference genome (version GRCm38.mm10) using the HISAT2 aligner (v2.1.0). Aligned reads were then counted using featurecount (v 1.6.2) to assign read counts to each annotation gene id. DESeq2 R Bioconductor package[46,47] was used to normalize read counts and identify differentially expressed (DE) genes. Kyoto Encyclopedia of Genes and Genomes (KEGG)[48] pathway data were downloaded using KEGG API (https://www.kegg.jp/kegg/rest/keggapi.html) and gene ontology (GO) data were downloaded from NCBI FTP (ftp://ftp.ncbi.nlm.nih.gov/gene/DATA/gene2go.gz). The enrichment of DE genes to pathways and GOs was calculated by Fisher's exact test in R statistical package. DE genes were determined using cutoffs of fold changes >2 and an adjusted $p$ value of <0.05.

**Chromatin accessibility analysis**. The ATAC-seq protocol was modified from prior published protocols[49]. Flash-frozen quadriceps muscle samples from P17 mice ($n = 3$ mice per genotype) were resuspended in 1 ml of homogenization buffer (5 mM CaCl₂, 3 mM Mg acetate, 10 mM Tris pH 7.8, 320 mM sucrose, 200 μM EDTA, 0.1% NP-40, 0.05% BME, with cOmplete protease inhibitor cocktail (Sigma, 11697498001)) and disrupted with beads in Precellys Evolution (3× 20 s at 6800 rpm). Lysate was sequentially filtered through 70- and 40-μm cell strainers, laid on top of sucrose buffer (1 M sucrose, 3 mM Mg acetate, 10 mM Tris pH 7.8) and centrifuged at $1000 \times g$ for 10 min. Nuclei in the pellet were permeabilized in 0.5 ml of 0.3% Triton in PBS for 30 min and washed twice with resuspension buffer (10 mM NaCl, 3 mM Mg acetate, 10 mM Tris pH 7.8). Transposition and library preparation were performed using TDE1 Tagment DNA Enzyme (Illumina, 15027865) and Nextera DNA Library Prep Kit (Illumina, 15027866) as per the manufacturer's instructions[49]. Sequencing ($n = 3$ mice per genotype) was performed by the UT Southwestern Genomic and Microarray Core Facility.

Paired-end raw reads were mapped to the mouse reference genome (GRCh38/mm10) using bowtie2 (version 2.3.4.3) with parameter "–very-sensitive" enabled. Read duplication and reads that mapped to chrM were removed from downstream analysis. Peaks were called using findpeaks command from the HOMER software package version 4.9, with parameter "–style dnase," and the false discovery rate (FDR) threshold (for Poisson $p$ value cutoff) was set to 0.001. Called peaks were merged from all samples and annotatePeaks.pl command was used to produce a raw count matrix. Differential peaks were identified using R package DEseq version 3.8. Differentially regulated peaks were determined using cutoffs of fold changes >2 and an adjusted $p$ value of <0.05. To analyze the functional significance of peaks, Genomic Regions Enrichment of Annotations Tool was used with mm10 as the background genome and other parameters set as default.

**Lamin A/C ChIP-seq**. Frozen hindlimb muscles ($n = 3$ mice per genotype) were crushed to powder and crosslinked in 10 ml of PBS with 2% formaldehyde (Sigma, F8775) for 15 min at room temperature under rotation. Crosslinking was stopped with 1.5 ml of 2.5 M glycine. Samples were washed with PBS and incubated on ice for 10 min in Farham lysis buffer (5 mM PIPES, 85 mM KCl, 0.5% N-40, pH 8.0) before bead lysis in Precellys Evolution (3× 20 s at 6800 rpm). Lysates were then incubated on ice for 20 min, and the supernatant was removed after centrifugation ($1000 \times g$, 5 min). The remaining pellet was resuspended in TE and 0.2% sodium dodecyl sulfate (SDS; 10 mM Tris-HCl pH 8.0, 1 mM EDTA), and nuclei were sonicated on Bioruptor pico (Diagenode) for 10 cycles (30 s on, 30 s off). In all, 1% of sheared DNA was saved for input, and the rest was diluted 1:1 with 1× TE 0.1% sodium deoxycholate and 1% Triton X-100 and 8 μg of Lamin A/C antibody was added (Santa Cruz, sc-7292 X) and conjugated to 60 μl of Protein G Dynabeads (ThermoFisher, 10003D). Lamin A/C immunoprecipitation was performed for 48 h at 4 °C, and beads were subsequently washed twice with RIPA buffer, 360 mM NaCl RIPA buffer, LiCl buffer (250 mM LiCl, 0.5% NP40, 0.5% deoxycholate, 1 mM EDTA, 10 mM Tris-HCl, pH 8.0), and TE buffer. DNA was released from the beads by addition of decrosslinking buffer (TE 0.3% SDS, 2 mg/ml proteinase K) and incubation at 65 °C for 16 h under constant mixing. RNA was then removed by incubation with 0.1 μg/μl of RNase A and incubation at 37 °C for 1 h under constant mixing. DNA was purified from the supernatant with the Qiagen PCR Purification Kit (Qiagen, 28104). Sequencing ($n = 3$ mice per genotype) was performed by the UT Southwestern Next Generation Sequencing Core.

Raw reads were mapped to the mouse reference genome (GRCh38/mm10) using bowtie2 (version 2.3.4.3) with default parameters. Duplicate reads were removed with "mark duplicates" from Picard tools (v.2.10.3). To detect LADs, Enriched Domain Detector (v.1.0) was used with a 10-Kb bin size, gap penalty of 10, and an FDR-adjusted significance threshold of 0.05. Gain, loss, and overlapping LADs between WT and KO samples were tallied using bedtools (v.2.29.0).

**Metabolomics**. Quadriceps from P17 WT and Net39 KO mice were harvested and flash-frozen in liquid nitrogen. Samples were homogenized in bead tubes with Precellys Evolution (3× 20 s at 6800 rpm) in 1 ml of methanol/water (80:20 vol/vol). In all, 200 μl of sample were transferred to a new tube with 800 μl of ice-cold methanol/water (80:20 vol/vol). Samples were vortexed for 1 min and centrifuged at $20,000 \times g$ for 15 min at 4 °C. Supernatant was transferred to a new tube and dried with SpeedVac system. Samples were further processed and analyzed as described here and in prior protocols[50]: samples were reconstituted in 0.03% formic acid,

vortexed, and debris was removed by centrifugation. The supernatant was used for the metabolomic studies. Liquid chromatography with tandem mass spectrometry (LC-MS/MS) was performed with AB QTRAP 5500 liquid chromatography–triple quadrupole mass spectrometer (Applied Biosystems SCIEX). Two mobile phases were used for separation: 0.03% formic acid in water and 0.03% formic acid in acetonitrile (ACN). MultiQuant software v.2.1 (Applied Biosystems SCIEX) was used to review the chromatogram and integrate peak area. The peak area for each metabolite was normalized to the total ion count of that sample. Metabolite identification targeted for 458 metabolites and 445 metabolites were detected above the baseline set by cell-free samples. Statistical differences were determined via partial least squares-discriminant analysis.

Limma R Bioconductor package[46,51] was used to identify differentially regulated pathways. KEGG[48] compound and pathway data were downloaded using KEGG API (https://www.kegg.jp/kegg/rest/keggapi.html). Differentially enriched pathways were determined by Fisher's exact test in R statistical package. Differentially regulated metabolites were determined using cutoffs of fold changes >2 and an adjusted $p$ value of <0.05. Raw data can be found in Supplementary Data 2.

Serum was collected from heart puncture, and glucose, insulin, triglycerides, cholesterol, and ketones were analyzed using VITROS clinical diagnostics.

**Mitochondrial DNA quantification**. Flash-frozen quadriceps muscle samples of P17 mice were homogenized in Trizol and phase-separated with chloroform. To the interphase and organic phase containing DNA, 4 M guanidine thiocyanate, 50 mM sodium citrate, and 1 M Tris were added; mixed; incubated at room temperature; and centrifuged at $3000 \times g$ at 4 C. The upper phase was transferred to a new tube and DNA was precipitated with isopropanol. DNA pellets were washed 4 times with 75% ethanol, resuspended in 8 mM NaOH, and HEPES and EDTA were added to a final concentration of 10 and 1 mM, respectively. The following primers were used for mtDNA qPCR (MT-MD1) and normalization (LPL):

NADH dehydrogenase subunit 1 (MT-ND1) Forward: 5′-CCCATTCGCGTTA TTCTT-3′ NADH dehydrogenase subunit 1 Reverse: 5′-AAGTTGATCGTAACGG AAGC-3′

LPL Forward: 5'-GGATGGACGGTAAGAGTGATTC-3'
LPL Reverse: 5'-ATCCAAGGGTAGCAGACAGGT-3'

**Proximity biotinylation in C2C12 cells**. Proximity biotinylation (BioID) was adapted from prior publications[52]. C2C12 myoblasts expressing pMXs-puro-Net39-miniTurbo were plated on 10 15-cm dishes at 100% confluence and differentiated in DM (DMEM with 2% horse serum and 1% antibiotic–antimycotic; ThermoFisher, 26050088) for 7 days. Five 5-cm dishes were supplemented with 500 μM biotin (Sigma, B4501) for 4 h. The remaining 5 15-cm dishes were used as negative control. Cell lysates were extracted in 1 ml of lysis buffer (6 M urea, 10% SDS, supplemented with cOmplete protease inhibitor cocktail, and PhosSTOP phosphatase inhibitor cocktail) and lysed mechanically with Precellys Evolution (3× 20 s at 6800 rpm). Lysates were added to 9 ml of dilution buffer (50 mM Tris, 150 mM NaCl) and 100 μl of equilibrated streptavidin magnetic beads (ThermoFisher, 88816). Lysates were incubated for 24 h at 4 °C on a wheel. Beads were washed 5 times with lysis buffer and boiled for 10 min in 2× Laemmli sample buffer (Bio-Rad, 1610737). Pulldown was assessed by silver staining (ThermoFisher, LC6070).

For protein identification by MS, samples were run for 1 cm in an Any-KD Mini-PROTEAN 10-well gel (Bio-Rad, # 4569034). Gels were then fixed and stained with EZBlue (Sigma, G1041) as per the provider's instructions. The area of the gel containing proteins was cut into small 1-mm cubes and submitted for analysis to the Proteomics Core Facility at University of Texas Southwestern Medical Center. Gel band samples were digested overnight with trypsin (Pierce) following reduction and alkylation with DTT and iodoacetamide (Sigma). The samples then underwent solid-phase extraction cleanup with an Oasis HLB plate (Waters), and the resulting samples were injected onto an Orbitrap Fusion Lumos mass spectrometer coupled to an Ultimate 3000 RSLC-Nano liquid chromatography system. Samples were injected onto a 75 μm i.d., 75-cm long EasySpray column (Thermofisher) and eluted with a gradient from 0 to 28% buffer B over 90 min. Buffer A contained 2% (v/v) ACN and 0.1% formic acid in water, and buffer B contained 80% (v/v) ACN, 10% (v/v) trifluoroethanol, and 0.1% formic acid in water. The mass spectrometer operated in positive ion mode with a source voltage of 1.8 kV and an ion transfer tube temperature of 275 °C. MS scans were acquired at 120,000 resolution in the Orbitrap and up to 10 MS/MS spectra were obtained in the ion trap for each full spectrum acquired using higher-energy collisional dissociation for ions with charges 2–7. Dynamic exclusion was set for 25 s after an ion was selected for fragmentation.

Raw MS data files were analyzed using Proteome Discoverer v2.2 (Thermofisher), with peptide identification performed using Sequest HT searching against the mouse protein database from UniProt along with the sequence for Net39-miniTurbo. Fragment and precursor tolerances of 10 ppm and 0.6 Da were specified, and three missed cleavages were allowed. Carbamidomethylation of Cys was set as a fixed modification, with oxidation of Met set as a variable modification. The FDR cutoff was 1% for all peptides. Two independent experiments for BioID were performed. Raw data for Net39 BioID can be found in Supplementary Data 1.

For analysis of enriched hits, results were filtered by enrichment (>20-fold enrichment in "Biotin" samples over "Control" samples) and ordered by

abundance. The top 50 highest hits were selected for analysis on STRING and the 5 most enriched GO terms were represented (Supplementary Fig. 5).

**Luciferase assays**. A region 442-bp upstream of the ORF of Net39 was used for promoter analysis based on MyoD ChIP-seq data on C2C12 differentiation (Supplementary Fig. 1a). Net39 promoter WT or with mutated E-boxes (Mut) were synthesized by Integrated DNA Technologies (IDT) and cloned into the promoterless luciferase reporter pGL4.10[luc2] (Promega, E6651) by conventional cloning. HEK 293T cells were transfected with combinations of reporter and either pCS2-GFP or pcDNA-MyoD-VP16. pcDNA-MyoD-VP16 encodes the bHLH domain of MyoD fused to the activation domain of VP16 and has been previously characterized[53]. All samples were transfected pCMV-LacZ for normalization of cell numbers. Forty-eight hours after transfection, luciferase assays were performed using Luciferase assay system (Promega, E1500) and beta-galactosidase assays were performed with the Mammalian beta-Galactosidase Assay Kit (Thermofisher, 75707) as per the provider's instructions. Luminescence and absorbance (405 nm) were read in a CLARIOstar plate reader (BMG Labtech).

**DNA primers**. A complete list of all primers used is included in Supplementary Data 3.

**Transthoracic echocardiography (ECHO)**. Cardiac function was determined by two-dimensional ECHO using the Visual Sonics Vevo 2100 Ultrasound (Visual Sonics, Toronto, ON, Canada) on conscious WT and Net39 KO mice at P9 and P17. Fractional shortening (FS) was calculated according to the following formula: $FS(\%) = [(LVID;d - LVID;s)/LVID;d] \times 100$. Left ventricular internal diameter (LVID) was measured as the largest anteroposterior diameter in either diastole (LVID;d) or systole (LVID;s). Ejection fraction (EF%) was calculated by: $EF(\%) = ([EDV - ESV]/EDV) \times 100$, where EDV is the end diastolic volume and ESV end systolic volume[54].

**Case selection and tissue processing**. The use of medical record and human tissues for research purposes was compliant with the ethical principles in the Belmont Report, the Department of Health and Human Service human subject regulations, Title 21 CFR, as well as good clinical practice (as adopted by the Food and Drug Administration), and approved by the UTSW Human Research Protection Program (IRB# STU012016-082). A waiver of patient informed consent was requested and approved by the human research protection program for retrospective study on archived human muscle tissue.

The pathology database at UTSW Medical Center was retrospectively reviewed. Among 10,070 muscle biopsies received between 1980 and 2016, 3 patients genetically confirmed to harbor Lamin A/C mutations and with available frozen muscle tissues were identified. Three normal muscle specimens from age-matched individuals served as controls. Muscle biopsies were collected from alive individuals and stored at −80 °C. Slides and electron microscopic images from all cases were reviewed by an experienced neuropathologist. Human muscle biopsy tissues were collected, processed, and analyzed according to all ethical regulations. Twenty-μm-thick cryosections from each muscle specimen were collected for western blot and quantitative reverse transcriptase PCR (qRT-PCR) analyses. The analysis of human muscle samples was performed by independent researchers. All tissues used in this study were coded and de-identified. Sample information is included in Table 1. RNA was isolated using the RNeasy Micro Kit (Qiagen, 74004) as per the provider's instructions. cDNA was synthesized using iScript Reverse Transcriptase (Bio-Rad). A Taqman probe (Hs00262043_m1, Thermofisher) was used for qRT-PCR analysis. Protein samples were extracted as described in prior sections.

**Statistics**. Data are presented as mean ± SEM. For histological and cellular experiments, statistical analysis was performed using one or two-tailed unpaired $t$ tests, as indicated in each figure legend. For genome-wide and metabolomics analysis, a fold change >2 and FDR <0.05 was used. Benjamini and Hochberg procedure was used for multiple hypothesis testing. Sample sizes and $p$ values are indicated in each figure legend.

**Reporting summary**. Further information on research design is available in the Nature Research Reporting Summary linked to this article.

## Data availability

All data needed to evaluate the conclusions in the paper are present in the paper and the Supplementary Data. Additional data related to this paper may be requested from the authors. Sequencing data were deposited in Gene Expression Omnibus (GEO) with the accession GSE154850. Source data are provided with this paper.

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

## Acknowledgements

We are grateful to J. Cabrera for graphics; R. Deberardinis for helpful discussion; and E. Sanchez-Ortiz, D. Karri, and L. Zacharias for technical assistance and reagents. We thank the Moody Foundation Flow Cytometry Facility for FACS, Children's Medical Center Research Institute Metabolomics and Sequencing Facilities for metabolomic analysis and next-generation sequencing, respectively, Genomics and Microarray core for RNA-seq, McDermott Next-Generation Sequencing Core for Lamin A/C ChIP library preparation and sequencing, Histo Pathology Core for histological analysis and expertise, UTSW Proteomics core for mass spectrometry, and UTSW Electron Microscopy Core Facility for expert technical assistance. This work was supported by funds from NIH (HL130253, AR071980 and AR-067294), the Senator Paul D. Wellstone Muscular Dystrophy Cooperative Research Center (U54 HD 087351), the Robert A. Welch Foundation (grant 1-0025 to E.N.O.), and the Canada Institute of Health Research Doctoral Foreign Study Award (DFS164267 to Y.Z.).

## Author contributions

A.R.-M., Y.Z., N.L., R.B.-D., and E.N.O. wrote and edited the manuscript. A.R.-M., B.K.C., and J.R.M. generated Net39 KO mice. A.R.-M. and Y.Z. performed histological, epigenetic, transcriptional and biochemical experiments. K.C., J.K, and L.X. performed bioinformatic analyses. C.C. obtained and evaluated human samples. J.M.S. and B.M.E. performed histological procedures and analysis. J.H. and P.P.A.M. performed electro-physiology measurements. A.B. performed western blot and image analysis.

## Competing interests

The authors declare no competing interests.
