## [Peer Review File · Nature Communications]

Reviewers' comments:

Reviewer #1 (Remarks to the Author):

This manuscript reports an exciting major advance in understanding the mechanisms of Emery-Dreifuss muscular dystrophy (EDMD). EDMD is caused by mutations that perturb the functioning of multiprotein complexes that span the nuclear envelope. Insights into this disease are essential to understanding how nuclear structure regulates genome architecture and tissue-specific control of the genome. Through extensive and clearly-described studies in a mouse knockout, the authors demonstrate fundamental and dramatic roles for the nuclear membrane protein NET39 in mouse survival, muscle physiology, sarcomere structure, muscle fiber type and muscle metabolism, as well as in the structural integrity of the nucleus, 3D genome organization and gene expression. NET39 also co-immunoprecipitates with three other nuclear membrane proteins linked to EDMD (Sun2, Lemd2, emerin), supporting the hypothesis that NET39 interacts functionally with EDMD-relevant proteins at the nuclear envelope.

Questions that must be addressed:

(1) Alternative explanations for the impact of lamin A mutations on NET39 must be considered. Figure 4D/E shows western blots of muscle biopsies from three LMNA-missense-mutated EDMD patients, and three controls, probed with antibodies against NET39. NET39 signals were reduced >80% in patients, and the authors concluded that 'Net39 expression is downregulated' (lines 35, 73 and 175). This important claim is not supported. The term 'downregulated' misleadingly implies 'transcriptionally downregulated', when mRNA levels/stability were not examined. More importantly, the authors must at least consider, and if possible also test or rule out, two plausible protein-level explanations for the apparent loss of NET39 protein in EDMD muscle: (a) enhanced turnover/degradation of wildtype NET39 protein, and (b) changes in posttranslational modifications that mask the epitope(s) recognized by this NET39 antibody [false-negative artifact].

(2) This manuscript does not report the BioID results. BioID data would be valued and interesting, but is not essential since the three candidates tested by co-IP (TAP purification) could have been predicted from known EDMD-relevant proteins. I suggest either (a) showing the BioID results, since they were used to determine which proteins were analyzed by Metascape GO/pathway enrichment, or (b) omitting the BioID results and deleting line 166 ("many of its potential binding partners identified by mass spectrometry are nuclear envelope proteins"), or (c) mention the BioID results and tell us they'll be published separately, if this is indeed the case.

(3) Briefly discuss the names and implications of the genes related to nuclear division, mitosis, and mitotic chromosome segregation that are downregulated in the KO in Day P17 (Fig 3E).

Line 34 implies EDMD is caused *solely* by mutations in lamin A. Must rephrase for accuracy, e.g.: "caused by mutations in A-type lamins (LMNA) and other genes".

Line 42-43: rephrase for accuracy, e.g., "...lamina, which includes intermediate filaments formed by lamins A, C, B1 and B2."

Lines 46-47: rephrase for accuracy, since nuclear pore complexes also mediate 'communication': "Mechanical communication between... through Links the Nucleoskeleton and Cytoskeleton (LINC) complexes, ..." [LINC complexes should always be plural, since there are so many types]

Line 53-60: simplify terms, and clarify that LADs are regions of DNA, e.g.: "In eukaryotes, transcriptionally active 'euchromatin' is typically found in the nuclear interior, whereas transcriptionally silent 'heterochromatin' adjoins the nuclear envelope [refs 8,9,10] and associates with nuclear lamins. Lamin-associated regions of DNA, termed lamin-associated domains (LADs), are dynamic and can redistribute upon gene activation.'

Line 64-66: Suggest simplification for accuracy (since there are two X-linked genes that cause EDMD), e.g.: "The two most frequent genetic causes of EDMD are X-linked recessive loss of emerin (encoded

by EMD) and autosomal dominant mutations in A-type lamins (encoded by LMNA) [refs 14,15]."

Lines 82-83: add meaning for non-experts, e.g., C2C12 myoblasts, Pax7+ primary myoblasts [what is the significance of Pax7?] and Tw2+ primary myoblasts [what is the significance of Tw2?].

Line 428: Reference 27 is missing from citations.

Line 444: Reference 28 is missing from citations.

Supplemental Figure 1C-- define TA, WAT and BAT.

Supplemental Figure 5C-- Stunning; move to Figure 2?

Supplemental Figure 7, line 750: add "(n = 3 mice each)".

Line 599: Is the difference in Figure 2D significant?

Figure 4B legend: State how many times this experiment (TAP purification of NET39-associated proteins) was done.

Figure 4D/E legend: State how many times this experiment was done, or whether quantification is based on the single experiment shown.

Reviewer #2 (Remarks to the Author):

In the manuscript entitled "Essentiality of the nuclear envelope protein Net39 for nuclear integrity, chromatin organization, and muscle growth" Ramirez-Martinez and colleagues characterized the striated muscle phenotypes and the survival of mice with deficiencies of Net39, a nuclear envelope protein. The manuscript is well written with straightforward conclusions from well-done experiments. However, I found the study rather descriptive by nature with several points that should be investigated in deeper details. The study stands short of really investigated the role of Net39 in skeletal muscle dysfunction

Major problems:

- the authors claim that loss of Net39 leads to EDMD-like phenotype. EDMD is a genetic condition characterized by early contractures, skeletal muscle weakness, and cardiomyopathy. It appears that Net39 is (almost) not expressed in cardiac tissue, therefore the cardiac function of the null mice was not investigated. I would suggest that the authors analyse in greater details the overall phenotype of the mice, given that they have a full knockout (and not conditional). Respiratory function should be addressed given that diaphragm displays expression of Net39.

- Given the interaction between Net39 and Sun/lemd2 and emerin, it could be extremely interesting to study this protein complex in normal and disease condition. The role in different cellular functions should be addressed in cellular and/or animal models.

-The author claim that the expression of Net39 is under the regulation by MoD and Myog. The experiments perform address this in part. It would be interesting to demonstrate how this regulation occurs (luciferase, promoter constructs,...). Without these, the claim remains only incomplete.

-The authors showed, using electron microscopy, that NE deformations are increased in Net39 KO mice. Without in vitro and in vivo rescue experiments, this remains an observation without link between the genetic loss and the cellular phenotype.

Minor problems:

- Introduction: "communication between the nucleus and the cytoplasm occurs through the Linker of Nucleoskeleton and the Cytoskeleton Complex". The Nuclear pore play a significant role allowing communication between nucleus and cytosol. This should clearly be said.

- Omics data should be available on public databases.

Reviewer #3 (Remarks to the Author):

The present study identified the transmembrane protein 39 (Net39) as a muscle-specific regulator of nuclear envelope activity and explores the role for this nuclear envelope protein in maintaining muscle chromatin organization.

Figure 1:

The evidence that Net39 is required for muscle structure and function relies on a global knockout mouse. Given their pronounced phenotype, additional phenotypic data on these knockout mice in addition to muscle tissue would be informative. For example, in Suppl Figure 6F, you demonstrate a marked reduction in serum glucose levels, please explain and include other metabolic parameters.

Figure 2:

Panel A - Is the picture of the nucleus from wild type and KO animals really taken at 20x as stated in the Figure legend? Showing multiple nuclei might be better to convey the key message. State statistical test used for analysis in Figure legend.

Panel D - statistical analysis is missing.

Figure 3:

State n-numbers used for ATAC-seq and RNA-seq in Figure legend.

What cut-offs were used for up- and down-regulated genes in KO muscle at P9 and P17?
How did you correct for multiple testing?

Additional analysis for lamin protein levels should be performed.

Figure 4.

Panel B - Validation of Net39 interaction with Sun2, Lemd2, and Emerin by co-immunoprecipitation was done in N2a cells, a mouse neuroblastoma cell line. Would other cell lines not be more appropriate?

Panel D - control and EMD samples should be run on same blot and include additional proteins, ie. lamins.

Panel E - Is vinculin a good loading control for nuclear envelopes? Can you include other proteins in addition to ensure equal loading?

mRNA expression and / or immunostaining of tissue biopsies from EDMD patients would be helpful to further confirm and elaborate on the findings in patients.

Suppl Figure 6.

Details on the metabolomics method are missing How many metabolites were screened and how were the metabolites identified?

Methods:

"Peptide identification and spectral counts were performed by the Proteomics Core Facility at University of Texas Southwestern Medical Center."

Experimental details should be included.

"Tandem affinity purification and proximity ligation proteomics analysis was integrated together. To filter noisy proteomics data, we plotted the distribution of logged FoldChange values and observed a bimodal distribution. The FoldChange threshold was chosen so that the peptides with low FoldChange value were filtered out from enrichment analysis."

Please be more specific. Where are these data deposited? What was the FoldChange threshold applied?

"Data are presented as mean \pm SEM. For histological and cellular experiments, statistical analysis was performed using two-tailed t-tests. For genome-wide and metabolomics analysis, a FDR >0.05 was used. Sample sizes and p-values are indicated in each figure."

The statistical paragraph is too short and lacks essential details.

Reviewers' comments:

Reviewer #1 (Remarks to the Author):

This manuscript reports an exciting major advance in understanding the mechanisms of Emery-Dreifuss muscular dystrophy (EDMD). EDMD is caused by mutations that perturb the functioning of multiprotein complexes that span the nuclear envelope. Insights into this disease are essential to understanding how nuclear structure regulates genome architecture and tissue-specific control of the genome. Through extensive and clearly-described studies in a mouse knockout, the authors demonstrate fundamental and dramatic roles for the nuclear membrane protein NET39 in mouse survival, muscle physiology, sarcomere structure, muscle fiber type and muscle metabolism, as well as in the structural integrity of the nucleus, 3D genome organization and gene expression. NET39 also co-immunoprecipitates with three other nuclear membrane proteins linked to EDMD (Sun2, Lemd2, emerin), supporting the hypothesis that NET39 interacts functionally with EDMD-relevant proteins at the nuclear envelope.

Response: We are grateful for the reviewer's enthusiasm and agree that our results could potentially help increase our understanding of a complex disease like Emery-Dreifuss muscular dystrophy. We appreciate the reviewer's insightful feedback.

Questions that must be addressed:

(1) Alternative explanations for the impact of lamin A mutations on NET39 must be considered. Figure 4D/E shows western blots of muscle biopsies from three LMNA-missense-mutated EDMD patients, and three controls, probed with antibodies against NET39. NET39 signals were reduced >80% in patients, and the authors concluded that 'Net39 expression is downregulated' (lines 35, 73 and 175). This important claim is not supported. The term 'downregulated' misleadingly implies 'transcriptionally downregulated', when mRNA levels/stability were not examined. More importantly, the authors must at least consider, and if possible also test or rule out, two plausible protein-level explanations for the apparent loss of NET39 protein in EDMD muscle: (a) enhanced turnover/degradation of wildtype NET39 protein, and (b) changes in posttranslational modifications that mask the epitope(s) recognized by this NET39 antibody [false-negative artifact].

Response: The reviewer highlights an important remark regarding the implications of "downregulated". To address this point, we have performed qRT-PCR analysis on those human biopsies to quantify the levels of Net39 transcript (revised Fig. 4d and revised Results, lines 202-204). We found that Net39 transcript is significantly downregulated, leading us to propose that reduced Net39 levels in EDMD patients are likely caused by reduced transcription.

To address the potential issue of the antibody epitope being masked, we tried using another antibody. The currently available antibodies against Net39 target the N-terminus (Abcam, Proteintech) or the C-terminus (Sigma). Regrettably, only the Sigma antibody

worked in our hands in these samples (including controls), so we do not have the proper tools to exclude the possibility of antibody masking.

Because of these above observations, we respectfully think that the markedly reduced expression of Net39 transcript in those patients is the most plausible explanation for the reduced Net39 protein levels (revised Fig. 4d and revised Results, lines 202-204).

(2) This manuscript does not report the BioID results. BioID data would be valued and interesting, but is not essential since the three candidates tested by co-IP (TAP purification) could have been predicted from known EDMD-relevant proteins. I suggest either (a) showing the BioID results, since they were used to determine which proteins were analyzed by Metascape GO/pathway enrichment, or (b) omitting the BioID results and deleting line 166 ("many of its potential binding partners identified by mass spectrometry are nuclear envelope proteins"), or (c) mention the BioID results and tell us they'll be published separately, if this is indeed the case.

Response: We apologize for not including a comprehensive list of the BioID results in the original manuscript. We have included the data showing all the proteins detected by BioID and their abundance (revised Supplementary Table 1).

(3) Briefly discuss the names and implications of the genes related to nuclear division, mitosis, and mitotic chromosome segregation that are downregulated in the KO in Day P17 (Fig 3E).

Response: Thank you! By addressing the reviewer's comments, we have improved the manuscript by discussing the genes involved in these pathways and have now addressed the implications of most of the pathways transcriptionally dysregulated in Net39 KO mice (revised Results, lines 169-171).

*Line 34 implies EDMD is caused *solely* by mutations in lamin A. Must rephrase for accuracy, e.g.: "caused by mutations in A-type lamins (LMNA) and other genes".*

Response: We apologize for this oversight and have made the suggested change (revised Abstract, line 37).

Line 42-43: rephrase for accuracy, e.g., "...lamina, which includes intermediate filaments formed by lamins A, C, B1 and B2."

Response: We have rephrased the sentence to be more accurate (revised Introduction, lines 46-47).

Lines 46-47: rephrase for accuracy, since nuclear pore complexes also mediate 'communication': "Mechanical communication between... through Links the Nucleoskeleton and Cytoskeleton (LINC) complexes, ..." [LINC complexes should always be plural, since there are so many types]

Response: We have rephrased the sentences and used the plural for LINC complexes (revised Introduction, lines 51-53).

Line 53-60: simplify terms, and clarify that LADs are regions of DNA, e.g.: "In eukaryotes, transcriptionally active 'euchromatin' is typically found in the nuclear interior, whereas transcriptionally silent 'heterochromatin' adjoins the nuclear envelope [refs 8,9,10] and associates with nuclear lamins. Lamin-associated regions of DNA, termed lamin-associated domains (LADs), are dynamic and can redistribute upon gene activation.'

Response: We have changed the sentence to make it clear and straightforward (revised Introduction, lines 58-62).

Line 64-66: Suggest simplification for accuracy (since there are two X-linked genes that cause EDMD), e.g.: "The two most frequent genetic causes of EDMD are X-linked recessive loss of emerin (encoded by EMD) and autosomal dominant mutations in A-type lamins (encoded by LMNA) [refs 14,15]."

Response: We have incorporated the reviewer's suggested changes (revised Introduction, lines 66-68).

Lines 82-83: add meaning for non-experts, e.g., C2C12 myoblasts, Pax7+ primary myoblasts [what is the significance of Pax7?] and Tw2+ primary myoblasts [what is the significance of Tw2?].

Response: We apologize for the lack of explanation. We have included additional text to explain the differences between the two muscle progenitor cells in terms of origin and localization (revised Results, lines 84-86).

Line 428: Reference 27 is missing from citations.

Line 444: Reference 28 is missing from citations.

Response: We reviewed the citations and resolved these issues. We apologize for the oversight.

Supplemental Figure 1C-- define TA, WAT and BAT.

Response: We have defined the acronyms in the figure legend.

Supplemental Figure 5C-- Stunning; move to Figure 2?

Response: We agree with the reviewer that the disarray in the diaphragm is striking and is likely the cause of death of the Net39 KO mice. Consequently, we moved the panel to the main figures (revised Fig.1g).

Supplemental Figure 7, line 750: add "(n = 3 mice each)".

Response: We have added “(n = 3 mice each)” to the figure legend.

Line 599: Is the difference in Figure 2D significant?

Response: While we observe a trend, the differences between KO EDL stretched and unstretched are not significant with n=3 (p=0.3). We have changed the text to reflect the lack of statistical significance and toned down our conclusion (revised Fig. 2d and revised Results, lines 125-127).

Figure 4B legend: State how many times this experiment (TAP purification of NET39-associated proteins) was done.

Response: We have performed TAP and BioID of Net39 twice (revised Methods, line 515). In the revised manuscript, we also included a third additional verification of BioID results by western blot analysis (revised Fig. 2f). Because Net39 is a nuclear membrane protein, which makes extraction and pulldown difficult, we believe that the results from BioID are more likely to reflect the actual interactome of Net39 and decided to exclude TAP proteomics data from the revised manuscript. To strengthen our findings, we validated the interaction between Net39 and endogenous Lemd2 by co-immunoprecipitation studies (revised Fig. 2g).

Figure 4D/E legend: State how many times this experiment was done, or whether quantification is based on the single experiment shown.

Response: The quantification was based on the single western blot shown in former Figure 4D. In this revised manuscript, we have repeated the EDMD western blots and used nuclear Histone H3 protein as a control for normalization (revised Fig. 4b, c) with similar results. We have stated in the figure legend that the quantification was done on the experiment shown.

Reviewer #2 (Remarks to the Author):

In the manuscript entitled “Essentiality of the nuclear envelope protein Net39 for nuclear integrity, chromatin organization, and muscle growth” Ramirez-Martinez and colleagues characterized the striated muscle phenotypes and the survival of mice with deficiencies of Net39, a nuclear envelope protein. The manuscript is well written with straightforward conclusions from well-done experiments. However, I found the study rather descriptive by nature with several points that should be investigated in deeper details. The study stands short of really investigated the role of Net39 in skeletal muscle dysfunction.

Response: We appreciate the reviewer’s comments on our manuscript. We agree that the original manuscript could be substantially improved with more mechanistic experiments. To address these concerns, we have included the following new data in the revised manuscript:

- 1) Validated Net39 interactions with other nuclear envelope proteins by BioID and co-immunoprecipitation in C2C12 cells (revised Fig. 2f, g). Intriguingly, one of the interacting partners (Lemd2) causes similar nuclear deformations when mutated in humans¹ (revised Results, lines 138-140).
- 2) Analyzed the protein composition of Net39 KO nuclear envelopes and observed increased protein levels in critical regulators of nuclear integrity (lamins) and Net39 binding partners (Lemd2) (revised Fig. 2e).
- 3) Examined localization of nuclear envelope proteins in Net39 KO muscle by immunofluorescence (revised Supplementary Fig. 5a).
- 4) Repeated EDMD western blot analysis, included an additional nuclear control, and showed that Net39 transcript is also downregulated in EDMD patients (revised Fig. 4b-d).
- 5) Performed luciferase assays to prove that Net39 is a MyoD target (revised Supplementary Fig. 1c).
- 6) Examined cardiac function in Net39 KO hearts (revised Supplementary Fig. 4).
- 7) Quantified the number of nuclei in Net39 KO muscles to assess hypoplasia (revised Supplementary Fig. 3).
- 8) Included additional blood serum measurements (revised Supplementary Fig. 9e).

These new data included in the revised manuscript provide mechanistic insights for our current study.

Major problems:

The authors claim that loss of Net39 leads to EDMD-like phenotype. EDMD is a genetic condition characterized by early contractures, skeletal muscle weakness, and cardiomyopathy. It appears that Net39 is (almost) not expressed in cardiac tissue, therefore the cardiac function of the null mice was not investigated. I would suggest that the authors analyse in greater details the overall phenotype of the mice, given that they have a full knockout and not conditional). Respiratory function should be addressed given that diaphragm displays expression of Net39.

Response: Because EDMD patients have a cardiac manifestation, we examined the hearts of the Net39 KO mice (revised Supplementary Fig. 4). We observed a reduction in cardiomyocyte size by histological analysis but no difference in cardiac function (ejection fraction and fractional shortening), as measured by echocardiography. Therefore, at P17 the hearts of Net39 KO mice do not show functional abnormalities. Mutations in other nuclear envelope genes manifest a late onset cardiomyopathy², and cardiac phenotypes in myopathic mice have been shown to manifest later in life³⁻⁴. Therefore, it is not surprising that Net39 KO mice do not show a functional cardiac phenotype at P17.

As for diaphragmatic function, we observed disorganized sarcomeres in Net39 KO diaphragms, and we moved this data to the main figures (revised Fig.1g) to emphasize its importance. Considering the ultrastructural abnormalities revealed by EM, diaphragm contractility is likely to be impaired. Regrettably, while our muscle stretching device allows for contractility measurements of some muscles like EDL and soleus, it cannot accommodate the unique shape of the diaphragm, so we have not been able to perform a direct contractility measurement. Similarly, respiratory function measurements like forced oscillation technique require ventricular cannulation, which, for us, has proven to be technically unfeasible for mouse pups of such a young age (17 days old), especially mice with a markedly runted phenotype like Net39 KO.

Given the interaction between Net39 and Sun/lemd2 and emerin, it could be extremely interesting to study this protein complex in normal and disease condition. The role in different cellular functions should be addressed in cellular and/or animal models.

Response: We appreciate the reviewer's enthusiasm and have performed additional experiments to study the Net39 complex with other proteins in normal and disease conditions while staying within the scope of the current manuscript.

To study Net39 interactions in a more physiological context, we have performed additional biochemical experiments in an *in vitro* muscle cell line, C2C12 myotubes. Our BioID findings were validated by an independent experiment and the hits were confirmed by western blot analysis (revised Fig. 2f). Importantly, we observed that in myotubes, Net39 associates with specific nuclear envelope proteins (Sun2, Lemd2, Emerin) but not others (Lamin A). This result provides mechanistic clues, as it implies that the changes observed by Lamin A ChIP-seq (revised Fig. 6a, b) are likely caused by the interaction of Net39 with proteins that regulate Lamin A activity, but not with Lamin A itself. To narrow down the list of candidate protein interactions, we validated them using additional assays such as co-immunoprecipitation in C2C12 myotubes. We detected endogenous Lemd2 in Net39 pulldown assays, providing more evidence for a direct interaction between Net39 and Lemd2 (revised Fig. 2g).

Additionally, we performed western blot analysis on Net39 KO skeletal muscle to assess the *in vivo* changes of proteins shown to be associated with Net39. Using Net39 KO skeletal muscle, we observed an increase in Lemd2 protein levels, but no changes with Emerin or Sun2 levels (revised Fig. 2e). Furthermore, Lamin A protein levels were also increased in Net39 KO. These changes may address the Net39 KO nuclear deformations,

since Lemd2 has been previously shown to tether DNA and A-type lamins to the envelope and be involved in regulating gene expression and the mechanical properties of the nucleus⁵. Interestingly, a homozygous point mutation in Lemd2 in humans also leads to nuclear envelope deformations in cardiomyocytes resembling those seen in Net39 KO myonuclei¹ (revised Results, lines 138-140, and revised Discussion, lines 214-218).

Altogether, our new data included to the revised manuscript provides further evidence suggesting that Net39 associates with Lemd2 and may regulate Lemd2 protein levels. These results suggest that Net39 associates with multiple components of the nuclear envelope and that their dysregulation compromises nuclear envelope integrity.

The authors claim that the expression of Net39 is under the regulation by MyoD and Myog. The experiments perform address this in part. It would be interesting to demonstrate how this regulation occurs (luciferase, promoter constructs...). Without these, the claim remains only incomplete.

Response: The revised manuscript includes luciferase assays with Net39 promoter as suggested (revised Supplementary Fig. 1c). We observed 12-fold induction of luciferase activity upon co-expression of MyoD. We then mutated MyoD binding sites (E-boxes) in the promoter, and MyoD co-expression did not increase luciferase activity. These results validate Net39 as a bona fide MyoD target.

The authors showed, using electron microscopy, that NE deformations are increased in Net39 KO mice. Without in vitro and in vivo rescue experiments, this remains an observation without link between the genetic loss and the cellular phenotype.

Response: We agree with the reviewer that our original manuscript would be enhanced by more data to assess the association between loss of Net39 and NE deformations. To better understand and describe the nature of the nuclear envelope deformations, we performed more experimental studies and included more data than just EM on KO nuclei in our revised manuscript. We measured the frequency of nuclear envelope deformations, and observed that they increase with age, consistent with increased activity and mechanical load (revised Fig. 2a). Furthermore, we experimentally augmented the phenotype by stretching WT and KO EDL muscles *ex vivo* and preparing those muscles for electron microscopy immediately after stretching. We found that 10 minutes of stretching was sufficient to induce a mild increase in nuclear deformations in the Net39 KO EDL muscle, but not in WT EDL muscles (revised Fig. 2d). These observations show that mechanical stress contributes to the severity of the phenotype of muscle lacking Net39. Supportive of our findings, it has been well-documented that nuclear deformations are associated with human envelopopathies (diseases with disruption of nuclear envelope components)⁶ and are not commonly seen with other muscle disorders⁷. This strongly suggests a direct link between the muscle specific nuclear envelope protein Net39 and the nuclear deformation phenotype in the Net39 KO mice. Overall, our data shows that Net39 is essential for the maintenance of nuclear envelope integrity, as well as, proper nuclear organization and gene expression.

Minor problems:

Introduction: "communication between the nucleus and the cytoplasm occurs through the Linker of Nucleoskeleton and the Cytoskeleton Complex". The Nuclear pore plays a significant role allowing communication between nucleus and cytosol. This should clearly be said.

Response: We apologize for omitting the importance of the nuclear pore in nucleus-cytosol crosstalk and have included it in the revised text (revised Introduction, lines 50-51).

Omics data should be available on public databases.

Response: We have uploaded the genomic datasets (ATAC-seq, RNA-seq, ChIP-seq) into public databases and are waiting for NCBI to provide an accession number, which will be included in the final manuscript. We included proteomic and metabolomics data as Supplementary tables (revised Supplementary Table 1 and 2).

Reviewer #3 (Remarks to the Author):

The present study identified the transmembrane protein 39 (Net39) as a muscle-specific regulator of nuclear envelope activity and explores the role for this nuclear envelope protein in maintaining muscle chromatin organization.

Response: We thank the reviewer for recommendations to improve our manuscript.

Figure 1:

The evidence that Net39 is required for muscle structure and function relies on a global knockout mouse. Given their pronounced phenotype, additional phenotypic data on these knockout mice in addition to muscle tissue would be informative. For example, in Suppl Figure 6F, you demonstrate a marked reduction in serum glucose levels, please explain and include other metabolic parameters.

Response: To address the reviewer's concerns, we expanded the panel of metabolic parameters examined in Net39 KO serum (revised Supplementary Fig. 8e). We observed a difference in serum glucose but no changes in other metabolites (triglycerides, cholesterol, ketones) or insulin. We included these findings in our revised manuscript (revised Results, lines 190-192). Consistent with our data, other reports^{8,9} describe muscle-specific deletion and overexpression of genes resulting in systemic changes in blood glucose. Since Net39 is a muscle-specific gene, we surmise that the changes in blood glucose are likely caused by alterations in muscle metabolism.

We agree with the reviewer's comment about including additional phenotypic data. We performed additional characterization of Net39 KO mice in other tissues. Gene expression analysis showed Net39 is restricted to muscle (revised Supplementary Fig.1). Because other nuclear envelope diseases manifest a cardiac phenotype (eg. Emery-Dreifuss muscular dystrophy), we examined Net39 KO hearts (revised Supplementary Fig.4). Histological analysis revealed a reduction in cardiomyocyte size, but echocardiography showed no difference in cardiac function (ejection fraction and fractional shortening) in Net39 KO mice. Therefore, cardiac function is not compromised in Net39 KO mice. Nevertheless, cardiac phenotypes can appear at later stages of life²⁻⁴. Since Net39 KO mouse die within 4 weeks of age, we are not able to measure functional cardiac changes that might develop with age.

Figure 2:

Panel A - Is the picture of the nucleus from wild type and KO animals really taken at 20x as stated in the Figure legend? Showing multiple nuclei might be better to convey the key message. State statistical test used for analysis in Figure legend.

Response: We apologize for the confusion. Revised Figure 2a has been updated to include the original 100x images on the left panel, and the magnified images showing multiple deformed nuclei on the right panel. We also indicated that the scale bar represents 20µm.

For statistical analysis, unpaired two-tailed t-tests were used to quantify differences between WT and KO images at each time point and muscle group (P4 QD, P9 QD, P17 QD, P17 GPS). This information has been included in the figure legend for clarity (revised Fig. 2a).

Panel D - statistical analysis is missing.

Response: We have included the statistical analysis for Fig. 2d in the figure legends: Unpaired, two-tailed t-tests were performed to quantify the differences between EDL unstretched and stretched images and between WT and KO images. While we observe a trend, with the current “n” the differences are not statistically significant. We have indicated that in the figure legends and toned-down our findings in the main text (revised Fig. 2d and revised Results, lines 125-127)

Figure 3:

State n-numbers used for ATAC-seq and RNA-seq in Figure legend.

Response: We included the number of replicates (n=3) in the figure legend (revised Fig. 3).

What cut-offs were used for up- and down-regulated genes in KO muscle at P9 and P17? How did you correct for multiple testing?

Response: We have fixed that omission in the revised manuscript. A fold change of greater than 2 and an adjusted p-value of less than 0.05 were used as the cutoff for the identification of differentially expressed genes at P9 and P17. This information has been included in the revised figure legends.

Additional analysis for lamin protein levels should be performed.

Response: We thank the reviewer for this suggestion. We performed western blot analysis of Lamin A and Lamin B1 protein in Net39 KO muscles (revised Fig. 2e). We observed increased levels of Lamin A and Lamin B1. We have included these data in our revised manuscript (revised Results, lines 128-131 and 148).

Figure 4.

Panel B - Validation of Net39 interaction with Sun2, Lemd2, and Emerin by co-immunoprecipitation was done in N2a cells, a mouse neuroblastoma cell line. Would other cell lines not be more appropriate?

Response: The reviewer raises an important point about studying the interaction between Net39 and other proteins in a more relevant cell line. In our revised manuscript, we performed additional biochemical assays to address that concern. We validated the mass spectrometry results from BioID with another streptavidin pulldown assay in C2C12 myotubes followed by western blot analysis (revised Fig. 2f) and we showed the

interaction between Net39 and endogenous Lemd2 by co-immunoprecipitation in C2C12 myotubes (revised Fig. 2g).

Panel D - control and EMD samples should be run on same blot and include additional proteins, ie. lamins.

Panel E - Is vinculin a good loading control for nuclear envelopes? Can you include other proteins in addition to ensure equal loading?

Response: We have performed additional western blot analysis on human biopsies to address the reviewer's comments. In our revised manuscript, all samples are on the same blot (revised Fig. 4b) and we included a nuclear protein (Histone H3) as a loading control. The revised densitometry (revised Fig. 4c) was normalized to Histone H3 giving similar results to the prior quantitation in the original manuscript.

We also analyzed Lamin A protein levels and observed no difference between control and EDMD patient samples (revised Fig. 4b, c). Although mutant Lamin A/C plays a dominant-negative role in autosomal dominant EDMD, it has been shown previously that Lamin protein levels do not necessarily differ between control and EDMD patients^{10,11}.

mRNA expression and / or immunostaining of tissue biopsies from EDMD patients would be helpful to further confirm and elaborate on the findings in patients.

Response: We performed qRT-PCR analysis of Net39 levels in muscle samples from healthy control and EDMD patients. We observed a decrease in Net39 mRNA transcript levels (revised Fig. 4d) and mentioned the results in the main text (revised Results, lines 202-204).

Suppl Figure 6.

Details on the metabolomics method are missing How many metabolites were screened and how were the metabolites identified?

Response: We included additional details on metabolomics procedures as per reviewer's request in the Method section (revised Methods, lines 448-472). Metabolite identification targeted for 458 metabolites, and 445 metabolites were detected above the baseline set by cell-free samples. We also included the measurements for all the metabolites detected (revised Supplementary Table 2).

Methods:

"Peptide identification and spectral counts were performed by the Proteomics Core Facility at University of Texas Southwestern Medical Center." Experimental details should be included.

Response: We expanded the Methods to include the experimental details of proteomic analysis (revised Methods, lines 490-517).

“Tandem affinity purification and proximity ligation proteomics analysis was integrated together. To filter noisy proteomics data, we plotted the distribution of logged FoldChange values and observed a bimodal distribution. The FoldChange threshold was chosen so that the peptides with low FoldChange value were filtered out from enrichment analysis.” Please be more specific. Where are these data deposited? What was the FoldChange threshold applied?

Response: Because Net39 is a nuclear membrane protein, extraction conditions compatible with pulldown can be challenging. Thus, we believe that the results from BioID are more likely to reflect the actual interactome of Net39. For clarity, we decided to exclude TAP proteomics data from the revised manuscript. We have redone the analysis of the BioID to reflect these changes and made the corresponding changes in the Methods to include the requested details (revised Methods, lines 518-521). We also included a supplementary table with all the BioID data (revised Supplementary Table 1).

“Data are presented as mean \pm SEM. For histological and cellular experiments, statistical analysis was performed using two-tailed *t*-tests. For genome-wide and metabolomics analysis, a FDR >0.05 was used. Sample sizes and *p*-values are indicated in each figure.” The statistical paragraph is too short and lacks essential details.

Response: Additional information on statistical analyses performed have been included in figure legends where statistical analyses were performed. For genome-wide, proteomics, and metabolomics analyses, additional statistical information has been included in the respective methods sections.

References

1. Abdelfatah, N. *et al.* Characterization of a Unique Form of Arrhythmic Cardiomyopathy Caused by Recessive Mutation in LEMD2. *JACC Basic to Transl. Sci.* **4**, 204–221 (2019).
2. Cattin, M. E. *et al.* Heterozygous Lmnad1K32 mice develop dilated cardiomyopathy through a combined pathomechanism of haploinsufficiency and peptide toxicity. *Hum. Mol. Genet.* **22**, 3152–3164 (2013).
3. Makarewich, C. A. *et al.* The DWORF micropeptide enhances contractility and prevents heart failure in a mouse model of dilated cardiomyopathy. *Elife* **7**, e38319 (2018).
4. Quinlan, J. G. *et al.* Evolution of the mdx mouse cardiomyopathy: Physiological and morphological findings. *Neuromuscul. Disord.* **14**, 491–496 (2004).
5. Schreiner, S. M., Koo, P. K., Zhao, Y., Mochrie, S. G. J. & King, M. C. The tethering of chromatin to the nuclear envelope supports nuclear mechanics. *Nat. Commun.* **6**, 7159 (2015).
6. Janin, A., Bauer, D., Ratti, F., Millat, G. & Méjat, A. Nuclear envelopopathies: a complex LINC between nuclear envelope and pathology. *Orphanet J. Rare Dis.* **12**, 147 (2017).
7. McNally, E. M. & Pytel, P. Muscle Diseases: The Muscular Dystrophies. *Annu. Rev. Pathol. Mech. Dis.* **2**, 87–109 (2007).

8. Li, L. O. *et al.* Compartmentalized Acyl-CoA metabolism in skeletal muscle regulates systemic glucose homeostasis. *Diabetes* **64**, 23–35 (2015).
9. Meng, Z. X. *et al.* Glucose Sensing by Skeletal Myocytes Couples Nutrient Signaling to Systemic Homeostasis. *Mol. Cell* **66**, 332-344.e4 (2017).
10. Reichart, B. *et al.* Expression and localization of nuclear proteins in autosomal-dominant Emery-Dreifuss muscular dystrophy with LMNA R377H mutation. *BMC Cell Biol.* **5**, 12 (2004).
11. Scharner, J. *et al.* Novel LMNA mutations in patients with Emery-Dreifuss muscular dystrophy and functional characterization of four LMNA mutations. *Hum. Mutat.* **32**, 152–167 (2011).

REVIEWER COMMENTS

Reviewer #1 (Remarks to the Author):

This revised manuscript shows convincingly that loss of NET39, a nuclear membrane protein, is profoundly disruptive in mice. Whole-animal NET39-KO mice are runted, with prominent muscle phenotypes including smaller muscle cells and disordered sarcomeres (Fig 1), perturbed nuclear structure both before and after muscle stretch (Fig 2), changes in 3D genome organization and the expression of genes relevant to muscle and metabolism (Fig 3), a shift from carbohydrate to lipid metabolism (Supp Fig 9), and increased resistance to fatigue (Supp Fig 8D). NET39-KO mice die within 25 days after birth (Fig. 1). Western blots of BioID proximity in cultured C2C12 myoblasts show NET39 associates directly or indirectly with three nuclear membrane proteins linked to Emery-Dreifuss muscular dystrophy (EDMD): emerin, LEMD2 and SUN2 (Fig 2F; Supp Table 1). The authors then test whether NET39 is perturbed in patients with dominant Emery-Dreifuss muscular dystrophy (EDMD)-causing mutations in LMNA, encoding A-type lamins. Western blots and RT-qPCR analysis of muscle biopsies from three EDMD patients with different mutations in LMNA all showed reduced expression of NET39 mRNA (Fig. 4D) and NET39 protein (Fig 4B/C). These results identify reduced NET39 as a contributor to the muscle defects in EDMD patients. These findings represent a significant body of work. However, several important issues must be addressed by revision.

(1) The authors must (a) fully summarize previous knowledge about NET39 (all three papers) in the introduction, and (b) discuss their own results accurately in relation to previous knowledge. This is vital for scholarship (to avoid false implication of 'first discovery'), vital for confirmation (where results in NET39-KO mice are consistent with studies in cultured cells), vital for discussion (where results differ), and vital to showcase results that are truly novel and understand their significance.

Related changes:

Lines 39-42: Delete or correct the last sentence of Abstract, which claims to reveal "an intimate role for the nuclear envelope in maintaining muscle chromatin organization, gene expression and function" (this role is well-established), and over-claims to reveal "the molecular etiology of EDMD".

Lines 136-138: Clarify that lamin A/C association (direct or indirect) with NET39 was expected from previous work (ref #18). Delete "direct physical" from line 137, because co-immunoprecipitation in Fig 2G does not prove direct binding, especially with nuclear lamina networks. Which LMNA spliceform (A? C? one major band on westerns) is expressed in mouse muscle?

The BioID proximity result in Fig2F is intriguingly suggests NET39 lacks proximity to lamin A/C; was this confirmed by a lack of enrichment for lamin A/C in Supp Table 1? 'Prelamin A' (lamin A and/or lamin C [any distinguishing peptides?]) is present in at least one raw dataset in Supp Table 1, but enrichment ratios are missing and there's no summary of overlap between the two datasets). Lack of proximity to lamin A/C for NET39, with four transmembrane domains, certainly motivates testing for potential proximity to nuclear membrane proteins. Is this manuscript the first to query NET39 association with other nuclear membrane or EDMD-relevant proteins (e.g., emerin, Sun1, Sun2, LEMD2, Lmo7, FHL1, nesprins)? Why do the authors focus on LEMD2? (Should delete Supp Fig 5a; uninformative because the LEMD2 antibody fluorescence signals are weak and mainly non-nuclear).

(2) The authors must properly analyze and discuss the BioID results, which were included as two raw datasets (Supp Table 1) with superficial STRING analysis (Supp Fig 5B). Provide a table with names of proteins enriched in both BioID experiments, show their enrichment ratios, discuss any that are EDMD-relevant, and discuss their enrichment relative to emerin (high co-IP signals with NET39 in Figure 2F) to assess NET39 association with other nuclear membrane protein(s) important in muscle (e.g., ref #1 and de Las Heras, 2017). Discuss the extent of overlap between the BioID results and the published NET39-co-IP proteome in ref #18, and whether mTOR or mTOR-associated proteins were identified by BioID.

(3) Must discuss the metabolomic and metabolic results in relation to NET39's published functional interaction with mTOR (ref #18). Do the authors' results support, conflict or provide new insight? Might a change in mTOR activity explain the reduced expression of mitosis-related genes in P17 knockout mice?

(4) Erroneous interpretation of Figure 2E (lines 130-131). This experiment was previously faulted because protein signals were normalized to vinculin, rather than a nuclear protein. This was not corrected in the revised manuscript.

The faulty interpretation: "by western blot analysis in NET39 KO muscles and found that LEMD2, LMNA and LMNB1 protein levels [ratio'd to vinculin] increased relative to WT muscle, whereas EMD and SUN2 protein levels did not change (Fig. 2e)."

The error: Western blot samples were from 17-day-old mice. Control muscle cells (full size) were compared to NET39-KO muscle, which have the same number of nuclei (Supp Fig 3) but are significantly smaller. The problem arises because gels were loaded with equal amounts of total protein (Methods, line 348), not equal numbers of nuclei. Hence, each NET39-KO lane contained (guesstimated) ~3-fold more nuclei than control lanes; this is the most plausible reason why NET39-KO muscle showed such high signals for LMNB1, LMNA and LEMD2 in Fig 2E, and further suggests that emerin and/or SUN2 protein levels might be reduced in NET39-KOs. If this possibility is pursued, the authors would need to test using two different antibodies to each protein (to avoid epitope-access artifacts).

The authors must either delete Figure 2E and all related text (e.g., line 148), or re-do this experiment, normalizing against histone H3.

(5) Supplemental Fig 8, showing a convincing shift in muscle fiber types, is beautiful and novel—consider moving to the main text?

(6) Compare the transcriptional changes in NET39-KO mouse muscle, with transcriptional changes in LMNA-mutated EDMD patients, which showed MyoD-dependent changes (Bakay M, Wang Z et al., 2006). E.g., if NET39 is regulated by MyoD and myogenin (lines 87-88), might this explain why NET39 transcription is reduced in LMNA-mutated patients?

(7) Briefly speculate why or how NET39 transcription would be affected by mutated LMNA.

Other changes and corrections:

Lines 68-69: name more EDMD-linked proteins, since several are NET39-proximal in the BioID data.

Lines 70-71 ("Here we identify the transmembrane protein 39 (Net39) as a muscle-specific regulator of nuclear envelope structure and function.") imply 'first discovery'—this is incorrect. This paragraph must be revised to accurately summarize previous NET39 results from three papers: Robson et al, 2016 (ref #1), Liu et al., 2009 (ref #18) and de Las Heras et al., 2017.

Line 74 and elsewhere: is three weeks after birth considered 'neonatal'?

Line 74: Can lethality be attributed solely to loss of NET39 function in muscle, when the entire animal lacks NET39?

Line 82 ("precise molecular functions within muscle in vivo are unknown") is incorrect. NET39 interacts functionally with mTOR (ref #18), and is known to tether silent chromatin in myogenic cells (ref #1 and de Las Heras 2017).

Lines 197-199: This sentence is inaccurate, overly dramatic and unnecessary: "Despite decades of research, it is still unclear why defects in ubiquitously-expressed lamins and other... EDMD".

Lines 199-200 ("The phenotypes of Net39 KO mice and EDMD mouse models are strikingly similar"): tone down or be more specific, because these mouse models are each different.

Line 230: "diminished in the nuclear envelope disease, EDMD" implies a simple disease; change to "diminished in EDMD patients with dominant LMNA missense mutations".

Line 235-237: "Our findings implicate Net39 in..." is incorrect, since NET39 was previously shown to regulate muscle-specific 3D chromatin organization. Rewrite to focus on the novel findings in this manuscript.

The Kubben reference is listed twice (references #27 and #29).

Line 748: change to "indicates the tail regions of lamin C and prelamin A."

Line 749: which human muscle was biopsied?

Lines 751 and 755: How many independent and/or technical repeats are represented?

Reviewer #2 (Remarks to the Author):

The authors did a great job at implementing their original work with well-designed new set of experiments. They answered most of my comments and I found the new version much more convincing.

Reviewer #4 (Remarks to the Author):

The manuscript entitled "The nuclear envelope protein Net39 is essential for nuclear integrity, chromatin organization, and muscle growth" reports on the function of the nuclear membrane protein NET39 which has an important role in sarcomere structure, muscle fiber type and metabolism. Furthermore, NET39 deficient nuclei are deformed and the loss of NET39 leads to altered gene expression, which leads to structural changes of the sarcomere and metabolic pathways. Overall, the manuscript is well-structured and all experiments are verified by ample controls. Moreover, the authors used a huge array of omics technologies to characterize the phenotype of NET39 deficient animals.

However, most of the results only describe the phenotype and a precise analysis of how NET39 induces changes in the nucleus and sarcomere remains unclear. For example, the authors demonstrated that the ablation of NET39 leads to a shift to enhanced oxidative activity. Unfortunately it remains unclear how this shift is associated to NET39. Do the authors observe more mitochondria? The validation of transcriptional co-regulators for mitochondrial biogenesis, such as PGC1alpha and mtDNA levels might help to clarify this observation.

In addition, the authors observed elevated levels of NADH, SDH and COX in the quadriceps muscle, which has a different fiber type composition and different amounts of mitochondria than the soleus and EDL muscle. Is it possible that certain fiber types and/or muscle groups that are subjected to greater stress are also more affected by NET39 ablation?

Could it be that the shift in fiber type composition is a result of a delayed growth and/or development of the mutants compared to controls? Although NET39 is not expressed in neurons, could it be that a dysfunctional innervation is responsible for the observed fiber type shift or muscle weakness?

The metabolomics analysis is interesting, but unfortunately does not help to decipher the function of NET39 in the nucleus. A correlation between enzymes involved in lipid metabolism and the increase in lipid species is an interesting finding, but does not contribute significantly to the functional characterization of NET39 in the nucleus. One might consider to separate this part from the current study.

The BioID approach is a superior technique for the identification of transient and weak interactors. Unfortunately, the number of background binders is usually quite high and this technique is not recommended for the identification of "new" and direct interactors. To demonstrate a direct protein-protein interaction, it would be better to perform "classical" affinity enrichment protocols, including an endogenous immunoprecipitation or the enrichment of tagged version of the NET39. However, since the authors substantiated the interaction of NET39 and LEMD2 by immunoblotting a further protein-protein interaction screen seems to be not necessary. However, it remains

unclear why a loss of NET39 results in increased levels of LEMD2 and why this effect is not observable in human Emery-Dreifuss muscular dystrophy patients. Given the increased LEMD2 concentrations, could it be that the free LEMD2 is more resistant to degradation compared to the NET39-LEM2 complex? Or could it be that the ablation of NET39 results in a transcriptional upregulation of LMD2?

Minor:

The SI table of the BioId dataset contains only intensity values without any ratios and statistical analysis. Although a comprehensive statistical analysis might be difficult since the authors performed only a duplicate analysis it might worse to show median ratios and p-values to better judge the quality of the dataset.

SI Figure 5b: Why have the authors used protein "domains" for the enrichment analysis. Is it possible to perform a 1D enrichment and report the overrepresented gene ontology terms? Please list the number of observed proteins hits (or domains) per category.

Reviewer 1

This revised manuscript shows convincingly that loss of NET39, a nuclear membrane protein, is profoundly disruptive in mice. Whole-animal NET39-KO mice are runted, with prominent muscle phenotypes including smaller muscle cells and disordered sarcomeres (Fig 1), perturbed nuclear structure both before and after muscle stretch (Fig 2), changes in 3D genome organization and the expression of genes relevant to muscle and metabolism (Fig 3), a shift from carbohydrate to lipid metabolism (Supp Fig 9), and increased resistance to fatigue (Supp Fig 8D). NET39-KO mice die within 25 days after birth (Fig. 1). Western blots of BioID proximity in cultured C2C12 myoblasts show NET39 associates directly or indirectly with three nuclear membrane proteins linked to Emery-Dreifuss muscular dystrophy (EDMD): emerin, LEMD2 and SUN2 (Fig 2F; Supp Table 1). The authors then test whether NET39 is perturbed in patients with dominant Emery-Dreifuss muscular dystrophy (EDMD)-causing mutations in LMNA, encoding A-type lamins. Western blots and RT-qPCR analysis of muscle biopsies from three EDMD patients with different mutations in LMNA all showed reduced expression of NET39 mRNA (Fig. 4D) and NET39 protein (Fig 4B/C). These results identify reduced NET39 as a contributor to the muscle defects in EDMD patients. These findings represent a significant body of work. However, several important issues must be addressed by revision.

Response: We thank the reviewer for the comments on the revised manuscript.

- (1) The authors must (a) fully summarize previous knowledge about NET39 (all three papers) in the introduction, and (b) discuss their own results accurately in relation to previous knowledge. This is vital for scholarship (to avoid false implication of 'first discovery'), vital for confirmation (where results in NET39-KO mice are consistent with studies in cultured cells), vital for discussion (where results differ), and vital to showcase results that are truly novel and understand their significance.

Response: We have expanded the manuscript to include the indicated changes.

- a) We included an additional paragraph in the Introduction summarizing the discovery of Net39, the three papers about its function in vitro and its potential link to human disease:

“Proteomic analysis of isolated nuclear envelopes²² has identified nuclear envelope transmembrane proteins (NETs) with potential links to human disease²³. Among them, transmembrane protein 39 (Net39), also referred to as inactive phospholipid phosphatase 7 (Plpp7 or Ppapdc3), has been studied in vitro with conflicting results^{1,24,25}. Net39 was initially reported to inhibit mTOR activity and IGF2 signaling, and knockdown of Net39 in C2C12 cells was shown to strongly promote myoblast differentiation²⁴. In contrast, later studies^{1,25} demonstrated that knockdown of Net39 blocked myogenesis and the main function of Net39 was to reposition specific genes that inhibit myoblast differentiation to the nuclear periphery, thus repressing their expression.” (lines 72-81)

- b) We discussed our results in relation to prior in vitro studies and elaborated on common findings, differences, explanation for conflicting results, and our new findings using an in vivo approach, as follows:

“In contrast to prior studies¹, we did not observe interaction between Net39 and lamins.”
(lines 154-155)

“While mTOR has been shown to be regulated by Net39 in transfected HeLa cells²⁴, no changes in mTOR signaling were observed in Net39 KO muscles (revised Supplementary Fig. 8g) and Net39 overexpression in C2C12 cells did not change mTOR localization (revised Supplementary Fig. 8h).” (lines 211-214)

“In vitro, Net39 has been described as both a negative and a positive regulator of C2C12 myoblast differentiation by regulating mTOR signaling and gene positioning, respectively^{1,24}. However, we did not observe changes in mTOR signaling in Net39 KO mice, or in C2C12 cells overexpressing Net39. It is possible that the previously reported functions may be specific to transfected HeLa cells and not observed in muscle cells. In contrast, Net39 KO mice presented extensive changes in genome organization that compromised muscle function. These observations are consistent with the other proposed function of Net39: to regulate myoblast differentiation by sequestering repressors of myogenesis to the nuclear periphery¹. (lines 257-265)

“Unbiased identification of Net39 interactors by proximity labeling (BioID) showed different results from those reported previously. By myc-Net39 pulldown, the nucleoplasmic N-terminus of Net39 was previously shown to interact with mTOR²⁴, and the N-terminus of Net39 has also been proposed to associate with Lamin A/C for genome tethering¹. Our proximity labeling data showed association of Net39 with EDMD-related proteins but not mTOR or any lamins. One explanation for the differences is that Net39 may control genome organization by interacting with lamin-associated proteins rather than lamins themselves. It is also possible that the interaction with lamins may be transient or too weak to be detected by BioID. Alternatively, the N-terminus of Net39 may not be accessible to the biotin ligase fused to the C-terminus of Net39 used in this study.”
(lines 266-276)

Related changes:

Lines 39-42: Delete or correct the last sentence of Abstract, which claims to reveal “an intimate role for the nuclear envelope in maintaining muscle chromatin organization, gene expression and function” (this role is well-established), and over-claims to reveal “the molecular etiology of EDMD”.

Response: The last sentence of the abstract has been revised as below (lines 39-42):

Original

Our findings reveal an intimate role for the nuclear envelope in maintaining muscle chromatin organization, gene expression and function, and highlight the importance of Net39 in these processes and in the molecular etiology of EDMD.

Revised

Our findings highlight the role of Net39 at the nuclear envelope in maintaining muscle chromatin organization, gene expression and function, and its potential contribution to the molecular etiology of EDMD.

Lines 136-138: Clarify that lamin A/C association (direct or indirect) with NET39 was expected from previous work (ref #18). Delete “direct physical” from line 137, because co-immunoprecipitation in Fig 2G does not prove direct binding, especially with nuclear lamina networks. Which LMNA spliceform (A? C? one major band on westerns) is expressed in mouse muscle?

Response: We indicated that the association with lamin was expected from prior studies and proposed three explanations for why we do not detect it by BioID (lines 267-276).

We deleted “direct physical”, as requested (line 150-151).

Both LMNA spliceforms are expressed in mouse muscle¹. The antibody we used for western blot analysis is specific for Lamin A spliceform (Abcam, ab26300).

The BioID proximity result in Fig2F is intriguingly suggests NET39 lacks proximity to lamin A/C; was this confirmed by a lack of enrichment for lamin A/C in Supp Table 1? ‘Prelamin A’ (lamin A and/or lamin C [any distinguishing peptides?]) is present in at least one raw dataset in Supp Table 1, but enrichment ratios are missing and there’s no summary of overlap between the two datasets).

Response: We included enrichment ratios in the revised Supplementary Table 1 and merged both datasets. While we detected lamin A/C in one round of BioID, it was not significantly enriched nor abundant. Validation of M/S by western blot analysis (Fig. 2f) confirmed the lack of proximity.

Lack of proximity to lamin A/C for NET39, with four transmembrane domains, certainly motivates testing for potential proximity to nuclear membrane proteins. Is this manuscript the first to query NET39 association with other nuclear membrane or EDMD-relevant proteins (e.g., emerin, Sun1, Sun2, LEMD2, Lmo7, FHL1, nesprins)?

Response: The first paper on Net39, by the Gerace group, performed myc-Net39 pulldown and identified lamin A/C. as an interacting protein. We discussed this in the revised manuscript (lines 267-276). To our knowledge, the association with other EDMD-relevant proteins has not been reported.

Why do the authors focus on LEMD2? (Should delete Supp Fig 5a; uninformative because the LEMD2 antibody fluorescence signals are weak and mainly non-nuclear).

Response: We focused on LEMD2 because:

1. We were able to validate the BioID result and confirm the interaction between NET39 and endogenous LEMD2 by co-IP in C2C12 cells (Fig. 2h and lines 155-156).
2. Net39 and Lemd2 may have similar functions in tethering genomic loci to the nuclear periphery² (lines 280-283).
3. A human Lemd2 mutation causes a nuclear deformation phenotype³ similar to that observed in Net39 KO mice (lines 156-159 and 278-280).

We deleted former Supplementary Figure 5a as suggested by the reviewer.

2. The authors must properly analyze and discuss the BioID results, which were included as two raw datasets (Supp Table 1) with superficial STRING analysis (Supp Fig 5B).

- a) Provide a table with names of proteins enriched in both BioID experiments, show their enrichment ratios

Response: We included enrichment ratios in the revised supplementary table and merged both datasets (revised Supplementary Table 1).

- b) discuss any that are EDMD-relevant, and discuss their enrichment relative to emerin (high co-IP signals with NET39 in Figure 2F)

Response: We originally validated BioID interactions by co-IP in N2a neuroblastoma cells (including emerin) but as suggested by reviewer #3, we re-did those experiments in a more relevant cell type. We were able to validate NET39 interaction with endogenous LEMD2 in C2C12 muscle cells by co-IP (Fig. 2h). Therefore, we focused most of our discussion on the implications of this interaction (lines 154-159 and 276-284).

- c) Assess NET39 association with other nuclear membrane protein(s) important in muscle (e.g., ref #1 and de Las Heras, 2017).

Response: We discussed the lack of association with LMNA by proximity labeling that would be expected from the indicated references (lines 154-155 and 269-276).

- d) Discuss the extent of overlap between the BioID results and the published NET39-co-IP proteome in ref #18, and whether mTOR or mTOR-associated proteins were identified by BioID.

Response: We discussed the previously reported interaction with mTOR and indicated it was not identified in our BioID experiments. We also discussed the limitations of each approach (lines 271-276). Furthermore, we performed western blot analysis showing no changes in mTOR signaling in Net39 KO muscles (revised Supplementary Fig. 8g and lines 211-212 and 259-261). The indicated paper from the Gerace group showed changes in mTOR localization when NET39 was overexpressed in HeLa cells, but we observed no

changes in mTOR localization in C2C12 cells overexpressing NET39 (revised Supplementary Fig. 8h and lines 213-214 and 259-261).

We respectfully consider that a more exhaustive in-depth comparison between BioID and published NET39 co-IP would not be informative. Both techniques are very different and NET39 co-IP identified not only mTOR, but other hits like GAPDH or tubulins that are likely contaminants when mild pull-down conditions, (like those used for co-IP) are used.

In short, we included the additional information for BioID, focused on a validated interaction relevant in the context of nuclear envelope biology, and discussed the discrepancies between our results and previously published interactions.

(3) Must discuss the metabolomic and metabolic results in relation to NET39's published functional interaction with mTOR (ref #18). Do the authors' results support, conflict or provide new insight? Might a change in mTOR activity explain the reduced expression of mitosis-related genes in P17 knockout mice?

Response: We observed no differences in mTOR signaling between WT and Net39 KO muscles (revised Supplementary Fig. 8g and lines 211-212 and 259-261). Furthermore, when NET39 was overexpressed in C2C12 myoblasts or myotubes, we did not observe changes in mTOR localization (revised Supplementary Fig. 8h and lines 213-214 and 259-261). These data suggest that changes in mTOR localization may occur in HeLa cells, as reported by ref#18, but not in other cell lines, such as C2C12 cells. We were not able to validate NET39-mTOR interaction by co-IP (data not shown).

(4) Erroneous interpretation of Figure 2E (lines 130-131). This experiment was previously faulted because protein signals were normalized to vinculin, rather than a nuclear protein. This was not corrected in the revised manuscript.

The faulty interpretation: "by western blot analysis in NET39 KO muscles and found that LEMD2, LMNA and LMNB1 protein levels [ratio'd to vinculin] increased relative to WT muscle, whereas EMD and SUN2 protein levels did not change (Fig. 2e)."

The error: Western blot samples were from 17-day-old mice. Control muscle cells (full size) were compared to NET39-KO muscle, which have the same number of nuclei (Supp Fig 3) but are significantly smaller. The problem arises because gels were loaded with equal amounts of total protein (Methods, line 348), not equal numbers of nuclei. Hence, each NET39-KO lane contained (guesstimated) ~3-fold more nuclei than control lanes; this is the most plausible reason why NET39-KO muscle showed such high signals for LMNB1, LMNA and LEMD2 in Fig 2E, and further suggests that emerin and/or SUN2 protein levels might be reduced in NET39-KOs. If this possibility is pursued, the authors would need to test using two different antibodies to each protein (to avoid epitope-access artifacts).

The authors must either delete Figure 2E and all related text (e.g., line 148), or re-do this experiment, normalizing against histone H3.

Response: In the current manuscript, we performed Histone H3 western blot analysis for WT and Net39 KO muscle (revised Fig. 2e). We then normalized the western blot signals

of nuclear envelope proteins and lamins against Histone H3 by densitometry (revised Fig. 2f). We observed a mild increase in H3 levels in Net39 KO muscles. H3 normalization showed changes in the relative levels of some nuclear envelope proteins and lamins (LMNB1, LEMD2, EMD) but not others (LMNA, SUN2) (lines 147-150).

We respectfully consider that a re-analysis of each protein with two different antibodies is unnecessary. There should not be epitope-access artifacts because:

1. Protein samples are denatured for western blot. Therefore epitope masking due to protein-protein interactions should not occur.
2. The epitopes are the same in WT and Net39 KO samples because there are no changes in the protein sequences of the proteins probed in these western blots.
3. The antibodies used for western blot analysis have been previously validated in many other publications in tissues or cells⁴⁻⁷.

(5) Supplemental Fig 8, showing a convincing shift in muscle fiber types, is beautiful and novel—consider moving to the main text?

Response: Thank you for the suggestion. However, we cannot move supplementary figures to the main text due to limits on the figure numbers.

(6) Compare the transcriptional changes in NET39-KO mouse muscle, with transcriptional changes in LMNA-mutated EDMD patients, which showed MyoD-dependent changes (Bakay M, Wang Z et al., 2006). E.g., if NET39 is regulated by MyoD and myogenin (lines 87-88), might this explain why NET39 transcription is reduced in LMNA-mutated patients?

Response: Thank you for the suggestion. We included this possibility in the Discussion and added the indicated reference (lines 302-303 and 307-309).

(7) Briefly speculate why or how NET39 transcription would be affected by mutated LMNA.

Response: As you suggested in point #6, it is possible that MyoD-dependent changes in LMNA mutants may contribute to NET39 transcriptional dysregulation. We have discussed this in the revised manuscript:

“Mutations in LMNA have been proposed to impair MyoD activation in the context of EDMD⁴³, and Net39 is a MyoD target (Supplementary Fig. 1c). Mutations in LMNA can also influence NET39 levels³⁷. We surmise that Lmna-dependent MyoD dysregulation could underlie the loss of the muscle-specific nuclear envelope protein Net39 in EDMD. Reduced Net39 levels may contribute to the muscle defects observed in EDMD.” (lines 302-307)

Other changes and corrections:

Lines 68-69: name more EDMD-linked proteins, since several are NET39-proximal in the BiOId data.

Response: Additional EDMD-linked genes were added to the Introduction: FHL1, TMEM43, Nesprins and Sun2 (lines 69-71).

Lines 70-71 (“Here we identify the transmembrane protein 39 (Net39) as a muscle-specific regulator of nuclear envelope structure and function.”) imply ‘first discovery’—this is incorrect. This paragraph must be revised to accurately summarize previous NET39 results from three papers: Robson et al, 2016 (ref #1), Liu et al., 2009 (ref #18) and de Las Heras et al., 2017.

Response: We included an additional paragraph in the Introduction summarizing the prior findings on Net39 *in vitro* (lines 72-81). We also changed the indicated statement to be more specific about the novelty of our manuscript (lines 82-83):

Original

Here we identify the transmembrane protein 39 (Net39) as a muscle-specific regulator of nuclear envelope structure and function.

Revised

*Here we explored the role of Net39 as a muscle-specific regulator of nuclear envelope structure *in vivo* by studying a Net39 knockout (KO) mouse model.*

Line 74 and elsewhere: is three weeks after birth considered ‘neonatal’?

Response: We changed the term from neonatal (up to 10 days) to juvenile (21-56 days old) for accuracy.

Line 74: Can lethality be attributed solely to loss of NET39 function in muscle, when the entire animal lacks NET39?

Response: Net39 expression is highly enriched in skeletal muscle (Supplementary Fig. 1d and lines 99-100) and the abnormalities in Net39 KO mice are restricted to skeletal muscle. Therefore, we respectfully consider that the lethality of Net39 KO mice can be attributed to impairment in muscle function.

Line 82 (“precise molecular functions within muscle *in vivo* are unknown”) is incorrect. NET39 interacts functionally with mTOR (ref #18), and is known to tether silent chromatin in myogenic cells (ref #1 and de Las Heras 2017).

Response: As we already indicated in the manuscript (lines 92-94): “*The muscle nuclear envelope protein Net39 has been shown to modulate myoblast differentiation **in vitro**^{1, 24}, but its precise molecular functions within muscle **in vivo** are unknown.*”

While the molecular function of Net39 has been studied *in vitro* (ie. muscle cell lines and HeLa cells), our study is the first to study the function of Net39 *in vivo* (ie. mouse model).

Lines 197-199: This sentence is inaccurate, overly dramatic and unnecessary: “Despite decades of research, it is still unclear why defects in ubiquitously-expressed lamins and other... EDMD”.

Response: We deleted the sentence from the manuscript.

Lines 199-200 (“The phenotypes of Net39 KO mice and EDMD mouse models are strikingly similar”): tone down or be more specific, because these mouse models are each different.

Response: We expanded that section of the manuscript to better explain what is known about Net39 in EDMD models and to be more specific about the overlap of the Net39 KO phenotype with a specific model of EDMD: *Lmna* Δ K32:

*“Several human mutations in NET39 have been identified as novel alleles associated with EDMD²³. EDMD is a complex disease and there are multiple mouse models that reflect its genetic diversity and broad range of manifestations⁷. The early lethality and nuclear envelope dysregulation observed in Net39 KO mice prompted us to compare the Net39 KO phenotype with other mouse models of severe EDMD. One such model carries a *Lmna* Δ K32 mutation, a single amino acid deletion that impairs the lateral assembly of lamin A/C³⁵. In humans, the *Lmna* Δ K32 mutation causes severe congenital muscular dystrophy³⁶. We observed overlapping phenotypes between *Lmna* Δ K32 mice³³ and our Net39 KO mice. Both mouse models manifest early lethality, nuclear abnormalities, failure to grow and metabolic alterations. Furthermore, it was recently reported that *Lmna* Δ K32 myotubes show downregulation of Net39 transcript and protein levels³⁷. These findings raised the possibility that Net39 expression may be affected in EDMD patients, and may contribute to the pathogenesis of the disease.”* (lines 226-238)

Line 230: “diminished in the nuclear envelope disease, EDMD” implies a simple disease; change to “diminished in EDMD patients with dominant LMNA missense mutations”.

Response: We incorporated the suggested change (lines 299-300).

Line 235-237: “Our findings implicate Net39 in...” is incorrect, since NET39 was previously shown to regulate muscle-specific 3D chromatin organization. Rewrite to focus on the novel findings in this manuscript.

Response: We changed the indicated sentence to better highlight the novelty of the manuscript (lines 309-311).

Original

Our findings implicate Net39 as a protein integral to muscle growth and function in mice and humans through its regulation of chromatin organization and nuclear envelope stability.

Revised

Overall, our findings show that loss of the nuclear envelope protein Net39 causes profound defects in mice, and the reduced NET39 levels in EDMD patients potentially contribute to the pathogenesis of this disorder.

The Kubben reference is listed twice (references #27 and #29).

Response: We corrected this oversight.

Line 748: change to “indicates the tail regions of lamin C and prelamin A.”

Response: This change is included in the revised manuscript (Fig. 4 and line 893-894).

Line 749: which human muscle was biopsied?

Response: Human biopsies were obtained from quadriceps, deltoid or thigh muscles. The specific information is included in the Methods section (lines 660-661).

Lines 751 and 755: How many independent and/or technical repeats are represented?

Response: The quantification is based on the western blot shown in Fig. 4b. We performed a prior western blot analysis and quantification in the first version of the manuscript with similar results for those samples. Therefore, each dot in Fig. 4c represents a different biological sample. This information is included in the figure legend (lines 896-898).

Reviewer 2

The authors did a great job at implementing their original work with well-designed new set of experiments. They answered most of my comments and I found the new version much more convincing.

Response: Thank you!

Reviewer 4

The manuscript entitled “The nuclear envelope protein Net39 is essential for nuclear integrity, chromatin organization, and muscle growth” reports on the function of the nuclear membrane protein NET39 which has an important role in sarcomere structure, muscle fiber type and metabolism. Furthermore, NET39 deficient nuclei are deformed and the loss of NET39 leads to altered gene expression, which leads to structural changes of the sarcomere and metabolic pathways. Overall, the manuscript is well-structured and all experiments are verified by ample controls. Moreover, the authors used a huge array of omics technologies to characterize the phenotype of NET39 deficient animals.

Response: We thank the reviewer for their time and comments to help improve our manuscript.

However, most of the results only describe the phenotype and a precise analysis of how NET39 induces changes in the nucleus and sarcomere remains mainly unclear. For example, the authors demonstrated that the ablation of NET39 leads to a shift to enhanced oxidative activity. Unfortunately, it remains unclear how this shift is associated to NET39. Do the authors observe more mitochondria? The validation of transcriptional co-regulators for mitochondrial biogenesis, such as PGC1alpha and mtDNA levels might help to clarify this observation.

Response: To assess mitochondrial biogenesis, we measured mtDNA content by qPCR analysis and observed no differences between WT and Net39 KO muscles (revised Supplementary Fig. 8e and lines 204-208). We observed no change in PGC1alpha transcript levels in Net39 KO muscle by RNA-seq (revised Supplementary Fig. 8f and lines 208-209). However, loss of Net39 leads to upregulation of transcripts involved in fatty acid metabolic processes and lipid oxidation (Fig. 3e and lines 183-185). Therefore, we conclude that the increased oxidative metabolism in Net39 KO muscle is not due to an increased number of mitochondria (lines 209-211 and 292-295).

In addition, the authors observed elevated levels of NADH, SDH and COX in the quadriceps muscle, which has a different fiber type composition and different amounts of mitochondria than the soleus and EDL muscle. Is it possible that certain fiber types and/or muscle groups that are subjected to greater stress are also more affected by NET39 ablation?

Response: It is true that different muscle groups are affected differently under pathological conditions. For instance, dystrophic muscles show differences in the extent of muscle damage, and exercise exacerbates the damage in some muscles (quadriceps) but not others (tibialis anterior)⁸.

In the case of Net39 KO mice, the changes observed in quadriceps for NADH, SDH and COX staining indicate increased oxidative metabolism (Supplementary Fig. 8c) and this observation is consistent with the increased fatigability and lower maximum tetanic forces measured in both EDL and soleus (Fig. 1f and Supplementary Fig. 8d). However, we

observed differences in the incidence of nuclear envelope deformations between quadriceps and gastrocnemius plantaris soleus (GPS) (Fig. 2a). Furthermore, RT-qPCR analysis of Net39 transcript shows a relative enrichment of Net39 expression in fast-twitch muscle types (such as tibialis anterior) over slow-twitch (soleus) (Supplementary Fig. 1d and lines 100-101). These data suggest that Net39 ablation may evoke subtle phenotypical differences among muscle types. Because the focus of our study is to understand the role of Net39 in vivo, we have included these observations in the manuscript:

“Generating a Net39 KO mouse model allowed us to understand the role of a tissue-specific nuclear envelope protein in vivo. Net39 expression is restricted to skeletal muscle, and deletion of Net39 caused more nuclear deformations in gastrocnemius plantaris soleus (GPS) muscles than in quadriceps muscle. Muscle groups can be differentially affected under pathological conditions such as muscular dystrophy⁴². We propose that heterogeneity in Net39 expression (enriched in fast-twitch muscle), mechanical burden, and fiber-type composition may account for subtle phenotypical differences across muscles.” (lines 285-292)

Could it be that the shift in fiber type composition is a result of a delayed growth and/or development of the mutants compared to controls? Although NET39 is not expressed in neurons, could it be that a dysfunctional innervation is responsible for the observed fiber type shift or muscle weakness?

Response: We do not think that the change in fiber type composition is a result of delayed growth and development of Net39 KO mice, because other mouse models with growth defects do not manifest changes in fiber-type or oxidative metabolism⁸.

We assessed innervation by immunofluorescence of neuronal axons (neurofilament, SV2) and acetylcholine receptor (bungarotoxin) in quadriceps to visualize the neuromuscular junction. We observed no differences between WT and KO muscles (Supplementary Fig. 3c). Thus, our data indicates that the decreases in muscle size and strength in Net39 KO mice are not caused by impaired innervation (lines 125-1301).

The metabolomics analysis is interesting, but unfortunately does not help to decipher the function of NET39 in the nucleus. A correlation between enzymes involved in lipid metabolism and the increase in lipid species is an interesting finding, but does not contribute significantly to the functional characterization of NET39 in the nucleus. One might consider to separate this part from the current study.

Response: Metabolic changes are observed in EDMD and other envelopopathies. For example, Lmna Δ K32 mice, which we discuss in the manuscript (lines 223-231), exhibit similar metabolic alterations⁹. We believe that the metabolomics data in Supplementary Fig.9, while not mechanistic, are of interest to better understand the overall phenotypic changes including fiber-type switch. Furthermore, these analyses can be compared with other models of EDMD (such as Lmna Δ K32 mice).

The BioID approach is a superior technique for the identification of transient and weak interactors. Unfortunately, the number of background binders is usually quite high and this technique is not recommended for the identification of “new” and direct interactors. To demonstrate a direct protein-protein interaction, it would be better to perform “classical” affinity enrichment protocols, including an endogenous immunoprecipitation or the enrichment of tagged version of the NET39. However, since the authors substantiated the interaction of NET39 and LEMD2 by immunoblotting a further protein-protein interaction screen seems to be not necessary.

Response: We validated the interaction between NET39 and endogenous LEMD2 in muscle cells by immunoprecipitation (Fig. 2h) and decided to focus on the potential implications of this interaction for this manuscript. We agree with the reviewer that other hits from BioID should be validated by additional techniques, but it will be more appropriate for future publications.

However, it remains unclear why a loss of NET39 results in increased levels of LEMD2 and why this effect is not observable in human Emery-Dreifuss muscular dystrophy patients. Given the increased LEMD2 concentrations, could it be that the free LEMD2 is more resistant to degradation compared to the NET39-LEMD2 complex? Or could it be that the ablation of NET39 results in a transcriptional upregulation of LMD2?

Response: RNA-seq in WT and Net39 KO quadriceps at P17 showed that Lemd2 is transcriptionally upregulated in Net39 KO muscle (lines 277-278). Net39 and Lemd2 have been proposed to have similar functions (lines 278-283). In the current manuscript, we hypothesize that Lemd2 upregulation may be a compensatory mechanism (lines 283-284).

We believe that the Net39 KO mouse model characterized in our manuscript represents a new and severe EDMD-like model. A detailed comparison with other mouse models and EDMD human datasets would be of interest in follow-up studies (lines 307-309).

Minor:

The SI table of the Biold dataset contains only intensity values without any ratios and statistical analysis. Although a comprehensive statistical analysis might be difficult since the authors performed only a duplicate analysis it might worse to show median ratios and p-values to better judge the quality of the dataset.

Response: We performed enrichment, and merged both replicates for a better assessment of the dataset. As the reviewer mentioned, because of the differences between experimental duplicates, we could not perform statistical analysis. (revised Supplementary Table 1).

SI Figure 5b: Why have the authors used protein “domains” for the enrichment analysis. Is it possible to perform a 1D enrichment and report the overrepresented gene ontology terms? Please list the number of observed proteins hits (or domains) per category.

Response: We have redone the analysis with the GO terms and included the observed proteins as indicated (revised Supplementary Fig. 5).

References

1. Xiong, L. et al. Linking skeletal muscle aging with osteoporosis by lamin A/C deficiency. *PLoS Biol.* 18(6): e3000731 (2020).
1. Brachner, A. et al. LEM2 is a novel MAN1-related inner nuclear membrane protein associated with A-type lamins. *J. Cell Sci.* **118**, 5797–5810 (2005).
2. Abdelfatah, N. et al. Characterization of a Unique Form of Arrhythmic Cardiomyopathy Caused by Recessive Mutation in LEMD2. *JACC Basic to Transl. Sci.* **4**, 204–221 (2019).
3. Jiao, X. et al. Dachshund Depletion Disrupts Mammary Gland Development and Diverts the Composition of the Mammary Gland Progenitor Pool. *Stem Cell Reports* **12**, 135–151 (2019).
4. Fichtman, B. et al. Combined loss of LAP1B and LAP1C results in an early onset multisystemic nuclear envelopathy. *Nat. Commun.* **10**, 605 (2019).
5. Huber, M. D., Guan, T. & Gerace, L. Overlapping Functions of Nuclear Envelope Proteins NET25 (Lem2) and Emerin in Regulation of Extracellular Signal-Regulated Kinase Signaling in Myoblast Differentiation. *Mol. Cell. Biol.* **29**, 5718–5728 (2009).
6. Ranade, D., Pradhan, R., Jayakrishnan, M., Hegde, S. & Sengupta, K. Lamin A/C and Emerin depletion impacts chromatin organization and dynamics in the interphase nucleus. *BMC Mol. Cell Biol.* **20**, 11 (2019).
7. Brussee, V. & Tremblay, J. P. Muscle fibers of mdx mice are more vulnerable to exercise than those of normal mice. *Neuromuscul. Disord.* **7**, 487–492 (1997).
8. Nakagawa, O. et al. Centronuclear myopathy in mice lacking a novel muscle-specific protein kinase transcriptionally regulated by MEF2. *Genes Dev.* **19**, 2066–2077 (2005).
9. Bertrand, A. T. et al. DelK32-lamin A/C has abnormal location and induces incomplete tissue maturation and severe metabolic defects leading to premature death. *Hum. Mol. Genet.* **21**, 1037–1048 (2012).

REVIEWERS' COMMENTS

Reviewer #1 (Remarks to the Author):

This revised manuscript is a major contribution (as noted previously) and beautifully addresses all but a few points, noted below, which can be corrected without re-review.

(1) Three text changes below are essential for accuracy, since: (i) "LMNA" is ambiguous (gene name; encodes both lamin A and lamin C); (ii) Abcam ab26300 recognizes lamin A only; and (iii) Lamin A and lamin C are functionally distinct (e.g.: Lamin C regulates genome organization after mitosis; doi: <https://doi.org/10.1101/2020.07.28.213884>).

Line 155: change "lamins" to "lamin A".

Line 448: Change "LMNA (Abcam, ab26300)" to "lamin A (Abcam, ab26300)".

Line 868: Change "LMNA" to "lamin A".

(2) Line 150 ("EMD") versus Line 154 ("Emerin") – use the protein name (Emerin) consistently.

(3) Results (line 150): Must add a brief caveat, since Net39-KO might alter posttranslational modifications of nuclear envelope proteins and thereby alter access to these epitopes on western blots. [Caveat is essential, since nuclear membrane proteins and lamins are extensively differentially regulated during interphase by phosphorylation and other PTMs, and the authors are relying on one antibody per target]

(4) Results (line 157): change "homozygous point" to "homozygous missense"

(5) Line 164: change "bound by" to "associated with"

(6) Discussion (line 280): change "Lemd2 tethers DNA" to "Lemd2 tethers chromatin"

Reviewer #4 (Remarks to the Author):

The authors have addressed all concerns of the reviewer and the reviewer has no further questions.

Reviewer #1

This revised manuscript is a major contribution (as noted previously) and beautifully addresses all but a few points, noted below, which can be corrected without re-review.

(1) Three text changes below are essential for accuracy, since: (i) “LMNA” is ambiguous (gene name; encodes both lamin A and lamin C); (ii) Abcam ab26300 recognizes lamin A only; and (iii) Lamin A and lamin C are functionally distinct (e.g.: Lamin C regulates genome organization after mitosis; doi: <https://doi.org/10.1101/2020.07.28.213884>).

Line 155: change “lamins” to “lamin A”.

Line 448: Change “LMNA (Abcam, ab26300)” to “lamin A (Abcam, ab26300)”.

Line 868: Change “LMNA” to “lamin A”.

Response: Thank you for your enthusiasm for our manuscript. We have included the indicated changes (lines 156, 452 and 892).

(2) Line 150 (“EMD”) versus Line 154 (“Emerin”) – use the protein name (Emerin) consistently.

Response: We have changed Emerin to EMD for consistency (line 150).

(3) Results (line 150): Must add a brief caveat, since Net39-KO might alter posttranslational modifications of nuclear envelope proteins and thereby alter access to these epitopes on western blots. [Caveat is essential, since nuclear membrane proteins and lamins are extensively differentially regulated during interphase by phosphorylation and other PTMs, and the authors are relying on one antibody per target]

Response: Yes, we see your point and have indicated in the manuscript that the changes detected by western blot could be caused by changes in protein levels or in post-translational modifications (lines 150-152).

(4) Results (line 157): change “homozygous point” to “homozygous missense”

(5) Line 164: change “bound by” to “associated with”

(6) Discussion (line 280): change “Lemd2 tethers DNA” to “Lemd2 tethers chromatin”

Response: We appreciate your corrections. We have included the indicated changes (line 158, 166-167 and 282).

Reviewer #4

The authors have addressed all concerns of the reviewer and the reviewer has no further questions.

Response: Thank you!